# Context-dependent agricultural intensification pathways to increase rice production in India

Hari Sankar Nayak ●[1] ✉, Andrew J. McDonald ●[1], Virender Kumar[2], Peter Craufurd[3], Shantanu Kumar Dubey[4], Amaresh Kumar Nayak[5], Chiter Mal Parihar[6], Panneerselvam Peramaiyan ●[7], Shishpal Poonia[8], Kindie Tesfaye ●[9], Ram K. Malik[8], Anton Urfels[1,2,10], Udham Singh Gautam[4] & João Vasco Silva ●[11]

Yield gap analysis is used to characterize the untapped production potential of cropping systems. With emerging large-*n* agronomic datasets and data science methods, pathways for narrowing yield gaps can be identified that provide actionable insights into where and how cropping systems can be sustainably intensified. Here we characterize the contributing factors to rice yield gaps across seven Indian states, with a case study region used to assess the power of intervention targeting. Primary yield constraints in the case study region were nitrogen and irrigation, but scenario analysis suggests modest average yield gains with universal adoption of higher nitrogen rates. When nitrogen limited fields are targeted for practice change (47% of the sample), yield gains are predicted to double. When nitrogen and irrigation co-limitations are targeted (20% of the sample), yield gains more than tripled. Results suggest that analytics-led strategies for crop intensification can generate transformative advances in productivity, profitability, and environmental outcomes.

Technologies and management strategies emerging from the 'Green Revolution' have transformed rice productivity and domestic food security across India since the 1970s[1]. Nevertheless, India's aggregate demand for rice is still projected to rise through mid-century[2,3], despite declining per capita consumption driven by demographic and dietary transitions[4]. India also contributes to global food security as world's largest rice exporter with a market share that exceeded 40% in 2022[5]. Greater demand for rice requires renewed efforts to understand where and how intensification (i.e., increasing rice yields per hectare) can be achieved without compromising environmental sustainability and farm-level profitability, which we refer to as sustainable intensification. Sustainable intensification is a particularly useful framework for rice cropping systems given their central importance to water resources, greenhouse gas emissions, and the livelihoods and food security of millions of smallholders[6–9]. The 2023 temporary export ban for certain types of rice also demonstrates the political importance of price stability and domestic rice production to the Government of India.

In most emerging economies, harnessing the benefits of crop genetic improvement while promoting 'best bet' packages of production practices remain the dominant model for agricultural

[1]School of Integrative Plant Science, Cornell University, Ithaca, NY, USA. [2]International Rice Research Institute (IRRI), Los Banos, Philippines. [3]International Maize and Wheat Improvement Center (CIMMYT), South Asia Regional Office, Lalitpur, Nepal. [4]Indian Council of Agricultural Research (ICAR), New Delhi, India. [5]ICAR-National Rice Research Institute, Cuttack, Odisha, India. [6]ICAR-Indian Agricultural Research Institute, New Delhi, India. [7]International Rice Research Institute (IRRI) - South Asia Regional Centre (ISARC), Varanasi, Uttar Pradesh, India. [8]International Maize and Wheat Improvement Center (CIMMYT), National Agricultural Science Complex (NASC), New Delhi, India. [9]International Maize and Wheat Improvement Center (CIMMYT), Addis Ababa, Ethiopia. [10]Water Resources Management Group, Wageningen University and Research, Wageningen, The Netherlands. [11]International Maize and Wheat Improvement Center (CIMMYT), Harrare, Zimbabwe. ✉e-mail: 1996harisankar@gmail.com; hsn28@cornell.edu

development[10,11]. Although most Indian farmers now plant modern cultivars[12], there is a wide variability in productivity outcomes for staple crops, even within individual states[13]. Smallholder-dominated farms in India have diverse soil properties, hydrological regimes, and crop management practices that together determine crop yields[14]. In sharp contrast to this heterogeneity, research-based crop management recommendations are primarily extrapolated from controlled-condition trials, representing a limited number of experimental sites, and commonly overlook interactions between production factors such as soil fertility management and irrigation practices[14]. As a complement to research station experiments, farmer research networks have emerged as an alternative approach for evaluating the context-dependent performance of agricultural innovations[15]. There are, however, open questions about the number of factors and interactions that can be studied through on-farm research as well as the financial and transaction costs of implementing these approaches across large areas.

Beyond on-station and on-farm research trials, recent advances in yield gap analysis offer another set of methodologies for characterizing the sustainable intensification potential of agroecosystems through observational data[16,17]. Nevertheless, for rice and other annual cereal crops, most existing studies rely on surveys with limited sample sizes[18] or, alternatively, crop models[19,20], remote sensing[21,22], or expert judgement[23] as the basis for analysis. More recently, Nayak et al.[24] used a comprehensive database of farmer field surveys to decompose (i.e., identify and characterize causal factors) rice yield gaps in Northwest India, but focused on population-level analysis without considering how yield constraints varied across fields and between sub-regions. Heterogeneity is an important consideration in complex production environments where smallholder farms predominate and diversity in crop management practices is common[25]. Recent advances in interpretable machine learning allows the identification of field-specific yield constraints and can support the development of site-specific insights and strategies for narrowing yield gaps through ex-ante scenario analysis[26,27].

The primary objectives of this study are two-fold: (i) to quantify the nature and factors contributing to attainable rice yield gaps in India, and (ii) to assess how analytics-based solutions may contribute to sustainable intensification through a case study in Bihar State and adjacent districts of Uttar Pradesh (hereafter referred to as 'Eastern India').

To address these objectives, we first aggregate a database of 15,876 field-year records for rice cultivation spanning seven major rice producing states in India (i.e., the LCAS - 'landscape crop assessment survey'). We then estimate the attainable yield gap for each state, defined, for our purposes, as the productivity difference between the top-yielding farmer fields (i.e., mean of the top 10% - attainable yield) and the mean of the remainder of the sample. With machine learning and diagnostic modelling, we also characterize the principal agronomic factors contributing to yield gaps in each state. Focusing on the attainable yield gap is a pragmatic choice since strategies for reaching biophysical yield potential are often not economically or environmentally desirable, whereas the attainable yield concept reflects agronomic management strategies that are already practiced in a region of interest, at least for some farmer segments[28].

We then develop a detailed case study of rice yield determinants in Eastern India based on the analysis of individual production fields. Eastern India is a priority development region for the Government of India that is endowed with a wealth of natural resources, particularly water, but has comparatively low crop productivity and rural incomes[29]. Our approach leverages a spatially balanced sampling framework and interpretation of machine learning yield predictions with SHapley Additive exPlanations (SHAP) values. We then use the same models to investigate the yield intensification and sustainability impacts of changes in key agronomic practices through ex-ante scenario analysis with and without solution targeting.

By taking advantage of large-$n$ surveys of crop yield and production practices, we hypothesize that machine learning combined with scenario and spatial analysis can identify context-dependent pathways for sustainable intensification. With this approach (hereafter referred to as 'analytics-based' methods), an overarching goal for this research is to develop an integrative methodology to identify where and how yield gaps can be narrowed while addressing multiple sustainable development objectives.

## Results

### Attainable rice yield gaps across Indian states

Average rice yield across the surveyed Indian states ranged between 3.3 t ha⁻¹ in Jharkhand and 5.5 t ha⁻¹ in Andhra Pradesh (Fig. 1a). Jharkhand also had the lowest attainable yield (5.1 t ha⁻¹) and Andhra Pradesh the highest at 7.7 t ha⁻¹. Despite notable differences in mean and attainable yield levels, yield gaps varied between 1.7 t ha⁻¹ in West Bengal to 2.4 t ha⁻¹ in Chhattisgarh (Fig. 1b). The median yield gaps in Bihar and Eastern Uttar Pradesh, Jharkhand, and Odisha were all estimated to be around 1.9 t ha⁻¹. The sizeable yield gaps documented in these data indicate considerable scope to increase rice production from existing land in India based on currently available technologies and management practices.

Random Forest (RF) models were developed to identify yield constraints for each state. RF explained between 29% (Odisha) and 52% (Andhra Pradesh) of the overall yield variation. Based on yield constraints, the two most important management practices for the attainable yield gap (Yg1 & Yg2) were identified and the potential to modify these practices to narrow the gap through improved agronomy was assessed for each state with individual conditional expectance (ICE) analysis (Fig. 2). Despite the perception that farmers generally overuse inputs in India, N fertilizer rate along with the number of irrigations were the main yield constraints in Odisha as well as in Bihar and Eastern Uttar Pradesh, accounting for an average anticipated yield gain of 0.2 t ha⁻¹ and 0.5 t ha⁻¹, in the respective states. These average values obscure the much higher yield gains anticipated in some fields. For example, improved management of N and irrigation are anticipated to boost yields by an average of 0.6 t ha⁻¹ in the most responsive fields in Odisha (i.e., top 25% of the yield gap distribution) and between 0.8 t ha⁻¹ and 2 t ha⁻¹ in Bihar and Eastern Uttar Pradesh. In West Bengal, K fertilizer emerged as the most important yield constraint, and rice variety and N fertilizer emerged in Jharkhand. In Chhattisgarh, insufficient N and P fertilizer rates were responsible for an average yield gain of 0.36 t ha⁻¹, with anticipated yield gains in the most responsive fields (i.e., top quartile) ranging between 0.5 and 2.0 t ha⁻¹. Finally, biophysical factors, not management practices, explained the largest share of the yield gap in Andhra Pradesh, hence the scope for rice yield increase through the top two management interventions, N and time of sowing, appears to be more limited than in other states with a combined yield gain of 0.27 t ha⁻¹.

### Determinants of rice productivity in Eastern India

The large number of observations ($n = 10,714$ field-year combinations) permitted a comprehensive analysis of rice yield constraints with the SHAP methodology for Eastern India. SHAP was used to quantify the impact of biophysical factors and management practices on predicted yield outcomes (i.e., decomposing attainable yield gaps) at the scale of individual production fields (Fig. 3). SHAP values can be interpreted as the yield deviation from the population mean (t ha⁻¹) attributable to a specific predictor for an individual production field. Number of irrigations, N, P, and Zn fertilizer rates emerged as the management practices with the largest influence on rice yield (Fig. 3A), whereas cumulative solar radiation and maximum temperature from sowing to harvest were the biophysical factors with the largest influence (Fig. 3B). For these specific data 'features', higher values were associated with higher SHAP values, hence higher rice yield prediction.

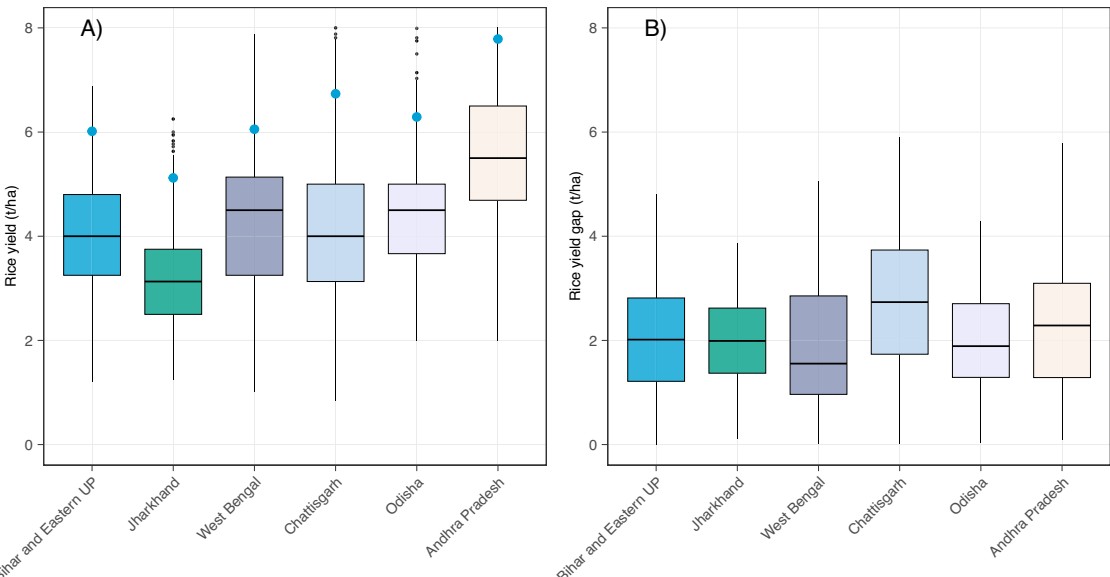

**Fig. 1 | Rice yields and yield gaps across seven states of India.** (**A**) Displays the rice yield variability and (**B**) the rice yield gap variability across field-year combinations in the dataset. Blue dots in (**A**) indicate the attainable yield for each state quantified as the mean yield of the top decile of production fields in each state. The yield gap distribution in (**B**) refers to the difference between the attainable yield and actual yields achieved in the remaining surveyed fields. The median yield gap value is depicted by the black line within each box and the inter-quartile range is defined by the box boundaries, whiskers extend up to 1.5 times inter quartile range. The number of field-year combinations in each state is as follow, Bihar and Eastern Uttar Pradesh ($n = 10,714$), Jharkhand ($n = 717$), West Bengal ($n = 1363$), Chhattisgarh ($n = 1099$), Odisha ($n = 747$), and Andhra Pradesh ($n = 1046$). Source data are provided as a Source Data file.

For features like sowing date (Julian date), larger values were associated with negative SHAP values, suggesting a negative impact of late planting on rice yield.

Hot spot analysis was used to generate spatial insights into the sub-regions of Eastern India where SHAP values for irrigation and N fertilizer were consistently high (red dots) or consistently low (blue dots) across all surveyed field-year combinations (Fig. 4). For N fertilizer, there were clear opportunity zones for increasing yields with higher N application rates particularly in the southeast and north-central regions of our case study region (Fig. 4A – points in blue). There were also areas where SHAP values were inconsistent across fields (i.e., denoted in grey) or areas where additional N use was not anticipated to translate into yield gains (i.e., points in red). For irrigation, even stronger spatial patterns emerged. Insufficient irrigation was predicted to limit rice yields in the southeast and northern half of the case study region (Fig. 4B – observations in blue), whereas areas in the south and southwest were less water-limited and unlikely to benefit from additional irrigation.

Next, to gain insight into the spatial extent and general distribution of co-limitations of N and irrigation, we clustered each surveyed year-field combination based on whether irrigation (I) and N were limiting (−; negative SHAP values) or less to non-limiting (+; positive SHAP values). This resulted in four clusters across the 10,714 field-year combinations out of which 35% where neither irrigation nor N was limiting ($I^+N^+$), 35% where irrigation was limiting ($I^-N^+$), 20% where both irrigation and N was limiting ($I^-N^-$), and 10% where only N was limiting ($I^+N^-$) (Supplementary Fig. 1). These results reflect the large number of fields limited by irrigation and N fertilizer rate, either individually or together. These clusters do not have a uniform spatial distribution, but all cluster types were found in every administrative district. For example, at the district level, 28–42% of the surveyed fields were not limited by N or by irrigation ($I^+N^+$) (Supplementary Fig. 2).

### Ex-ante evaluation of targeted versus 'blanket' management strategies in Eastern India

After constructing a yield model and characterizing the key drivers of yield outcomes, we then compared different strategies for achieving sustainable rice intensification through a scenario analysis. Our goal was to evaluate how targeted recommendations, if adopted, compare with uniform ('blanket') recommendations where all farms adopt the same management practice. Input use, predicted yield, and profitability were assessed for each scenario and aggregated across the region. For Scenario 1, all fields received an N rate of 125 kg N ha$^{-1}$ (i.e., current state recommendation). For Scenario 2, a blanket N rate of 180 kg N ha$^{-1}$ was assessed; this rate represents the population-level non-limiting rate for Eastern India emerging from our analysis (see Methods). Scenario 3 uses a targeting approach to adjust the N rate for only those fields with a negative SHAP value for N to a rate of 180 kg N ha$^{-1}$. Scenario 4 changes N rate to 180 kg ha$^{-1}$ and irrigation rate to 5 for fields where both factors were predicted by SHAP to limit rice yield (i.e., negative SHAP values for both factors). The last scenario was designed to evaluate how the geography of opportunity shifts when multiple interventions are considered.

Blanket use of the existing state N recommendation (Scenario 1) was predicted to produce an additional 0.15 million tons of rice in Eastern India with a small decrease of 0.016 million tons in total N use compared to current farmer practice (Supplementary Table 1). At the field scale, the implications of this strategy were highly variable with farmers in some districts anticipated to have very significant yield losses (*data not shown*). Blanket use of an analytics-based N recommendation (180 kg N ha$^{-1}$) for all rice fields (Scenario 2) resulted in an increase in rice production of 0.58 million tons but with an additional 0.22 million tons of N use as compared to the current farmer practice. Conversely, using a targeted approach to N management (Scenario 3) resulted in N increases to 180 kg ha$^{-1}$ for only 47% of all fields, producing an estimated additional rice production of 0.41 million tons while using an additional 0.13 million tons of N. In comparison to Scenario 2, this outcome represents a 21% gain in N use efficiency ('NUE' defined as kg grain kg N$^{-1}$) associated with additional N use above current farmer practice. In other words, the opportunity targeting in Scenario 3 appears to offer transformative gains in NUE over yield intensification strategies that use analytics to define an optimal N rate at the population level.

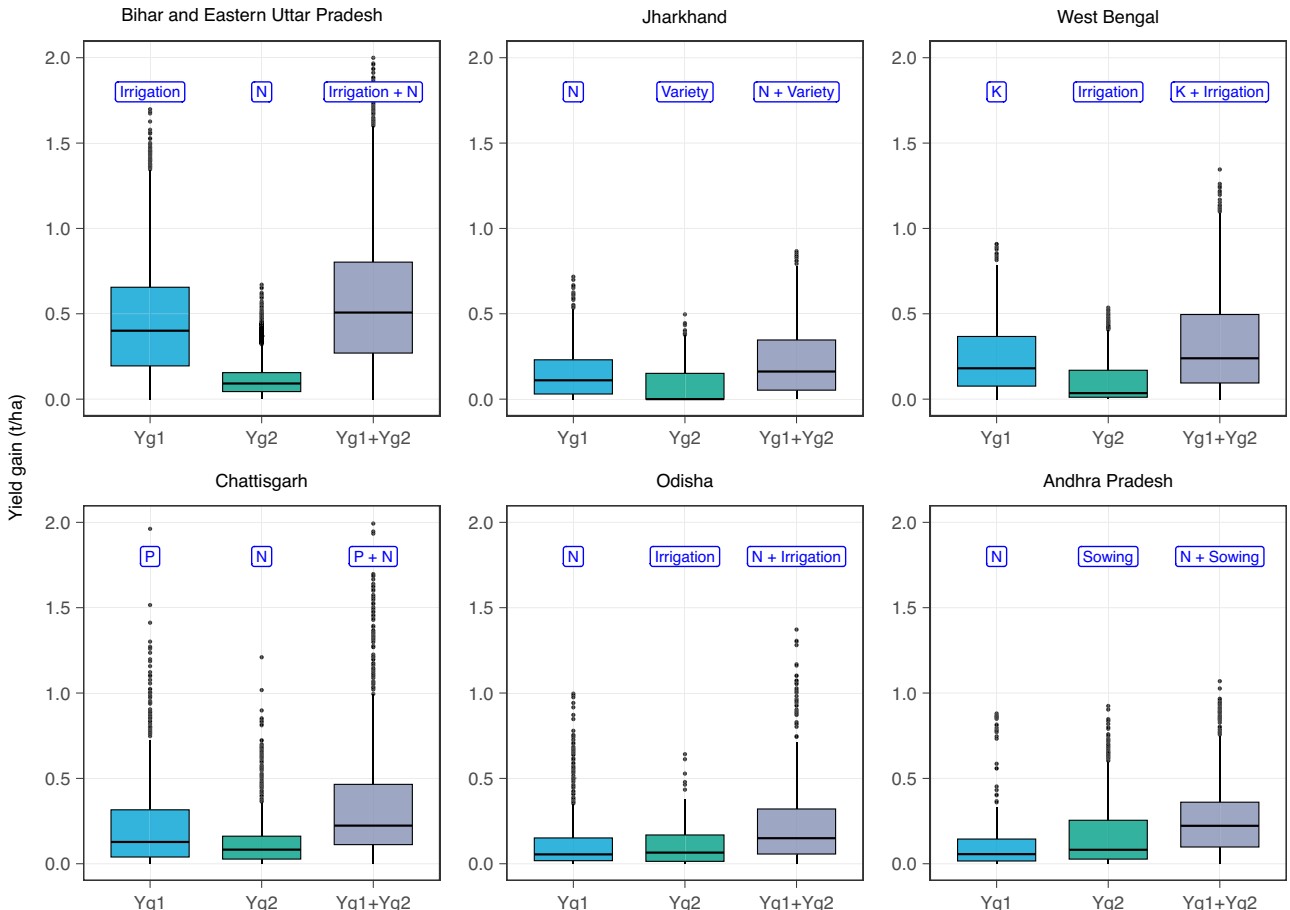

**Fig. 2 | Yield gain associated with the two most important management practices in seven states of Eastern India.** Potential yield gains (t ha⁻¹) associated with improved management of the top two most important agronomic constraints (Yg1, Yg2) for each India state as estimated by individual conditional expectance (ICE) analysis. Boxplots represent the distribution of yield gains predicted at the scale of individual farm fields. The third boxplot in each panel represents the combined effect of addressing both yield constraints (Yg1 + Yg2). Boxplot shows the median values and inter-quartile range as defined by the boxplot boundaries for Indian states of Bihar and Eastern Uttar Pradesh ($n$ = 10,714), Jharkhand ($n$ = 717), West Bengal ($n$ = 1363), Chhattisgarh ($n$ = 1099), Odisha ($n$ = 747), and Andhra Pradesh ($n$ = 1046). The whisker extends up to 1.5 times interquartile range and data points beyond these whiskers are represented as individual points. Source data are provided as a Source Data file.

By targeting fields where yields were co-limited by N and irrigation (Scenario 4), practice changes were only implemented in 20% of all rice fields. Among this sub-population, simultaneous changes to N and irrigation management were predicted to produce an additional 0.56 million tons of rice with a modest investment of 0.08 million tons of N in combination with an increase in irrigation that ranged from 1 to 4 events per field, a change that scales at the regional level to 2.33 million additional irrigations per season, approximately equivalent to an average of 17% of the water safely available for future use, which varies across the districts[30] (Supplementary Table 1). It's important to note that district-wise predicted changes for each scenario varied (Supplementary Fig. 3).

Changes in predicted rice yield and profitability (defined as partial net returns) at the district level varied within and across scenarios to varying degrees (Fig. 5). With a uniform N rate of 180 kg N ha⁻¹ applied to all fields (Scenario 2), the average yield gain over current farm practices was 0.15 t ha⁻¹ (i.e., 3.5% increase), with a profit gain of US$26 ha⁻¹ (Fig. 5A, B). With a targeted approach to N management based on SHAP values (Scenario 3, 47% of the study region), average yield gains doubled to 0.31 t ha⁻¹ and profit gains to about US $60 ha⁻¹, with the largest gains predicted in the eastern part of our case study region (Fig. 5C, D). In Scenario 4, where only fields with a co-limitation of N and irrigation were targeted (20% of the case study region), the average predicted yield gain over existing farm practices was 0.68 t ha⁻¹ (i.e., 19% increase) with a profit gain of $90 USD ha⁻¹ (Fig. 5E, F).

## Discussion

Rice is a dietary staple for more than 3.5 billion people, and in India serves as the primary foundation for food security and as an important export crop[5,31–33]. Moreover, demand for rice in India is anticipated to rise by as much as 50% by mid-century due to population increases[34,35]. Meeting these needs is increasingly challenging because of groundwater depletion, soil degradation, environmental pollution, and declining input use inefficiency that jeopardize sustainability goals in different parts of South Asia[7,36,37].

The Government of India has focused on rice intensification in regions where yield gaps are perceived to be high (e.g., 'Bringing the Green Revolution to Eastern India')[38]. States in Eastern India are considered rich in water resources but are typified by low productivity and low farm income[29]. These areas stand in contrast to Northwest India – known as the country's breadbasket where the scope to increase rice productivity is small[24]. Nevertheless, insights into the nature of yield gaps in the emerging priority regions are generalized and incompletely understood. In this study, we used an analytics-based approach to fill this knowledge gap and identify context-dependent pathways for sustainable rice intensification. Our approach quantifies field-specific yield constraints, in contrast to earlier studies providing population

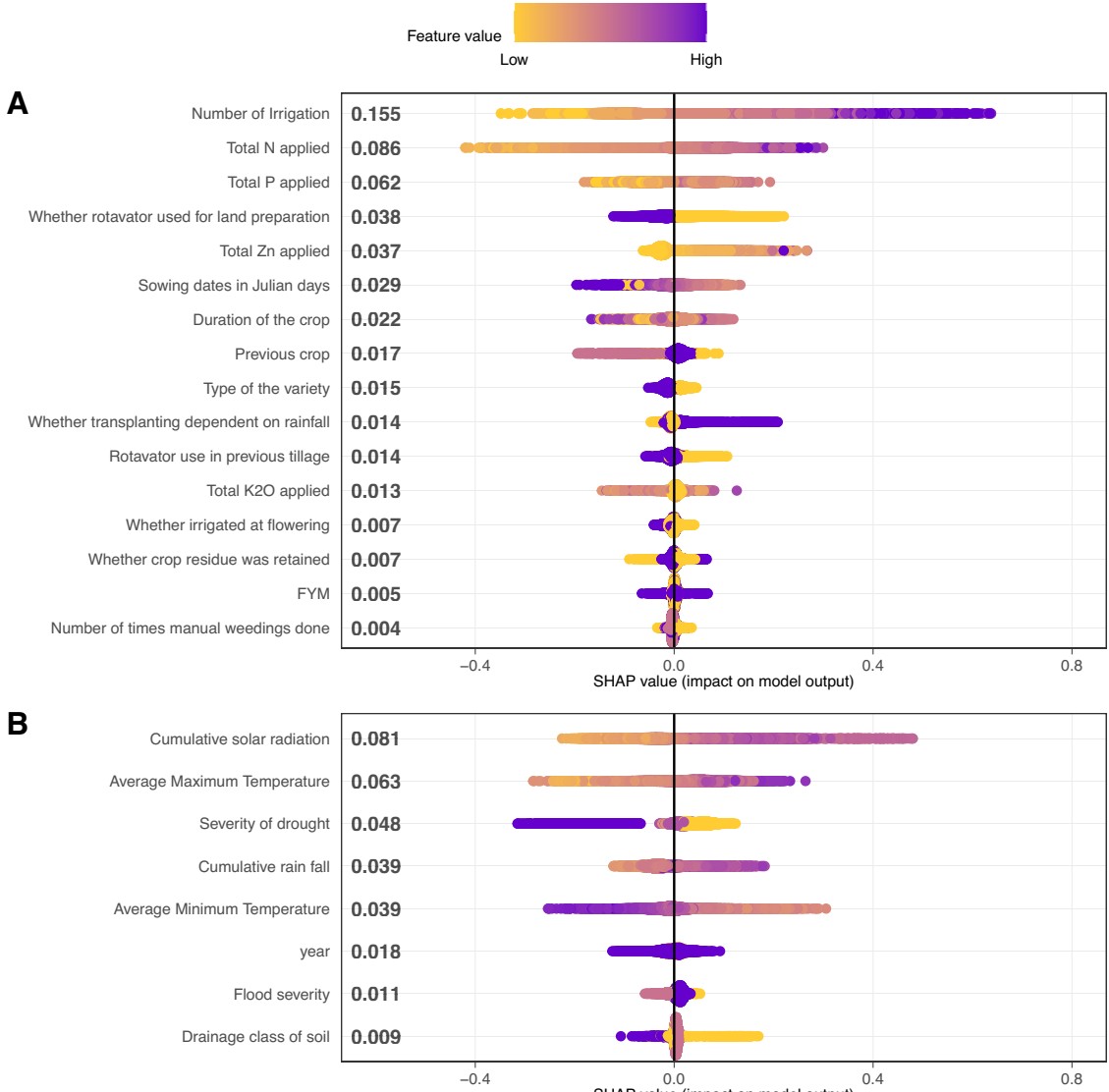

**Fig. 3 | SHAP summary plot of yield constraints in farmers' fields across Eastern India.** Drivers of rice yield in Eastern India as estimated by SHapely Additive exPlanation (SHAP) values, a post hoc method for assessing contributions of each feature to random forest crop yield predictions. Distributions of SHAP values from individual fields were grouped into management practices (**A**) and biophysical attributes (**B**). The color ramp indicates the actual value for the numeric variables or the assigned value (nominal) for categorical variables (see Supplementary Table 2 for further information). Variable importance was estimated by calculating the mean absolute SHAP value for all variables and is reported as numeric values along the y-axis. Source data are provided as a Source Data file.

level yield gap insights[14,39,40]. This type of local interpretation of machine learning models can support intervention targeting to more effectively and efficiently narrow yield gaps in production fields that are likely to accrue the highest benefits.

At the scale of different production regions, average attainable yield gaps for rice ranged between 1.8 and 2.8 t ha⁻¹. This means that the average field achieves 55–65% of the rice yields obtained by top performing farms in each state, suggesting considerable scope to increase productivity with existing technologies. This identified scope for rice yield improvement is lower than in global assessments that consider biophysical potential yield as a benchmark to estimate yield gaps (see Yuan et al.[23]), but also more pragmatic since the benchmarks are not theoretical. The main agronomic factors contributing to yield gaps differed by state and included irrigation management, fertilizer application rates (N and phosphorous, and zinc), variety maturity class, and transplanting dates. Interestingly, only some of these factors feature in the Government of India's investment strategies for rice intensification in lower-yielding production environments[38], and

constraints like insufficient N fertilizer are at odds with public policies and environmental concerns that seek broad-based reductions in N use in agriculture[41].

Distinct development priorities emerged for each of our studied regions, but this does not imply a 'one size fits all' approach to rice intensification given the heterogeneity of management and environmental factors within each region. To determine the potential importance of a targeted and analytics-based approach, we developed a machine learning model for rice yield to predict productivity outcomes under hypothetical scenarios of change for N and irrigation management in Bihar and Eastern Uttar Pradesh states of Eastern India, a case study region where these two factors were the top contributors to attainable yield gaps (Fig. 2) with a strong spatial dependence (Fig. 4).

Encouraging farmers to use the state-recommended 'blanket' N rate is an ostensibly simple approach for avoiding over- or under-use of fertilizer. But such strategy does not lead to significant production gains in our case study region in Eastern India. On the other hand,

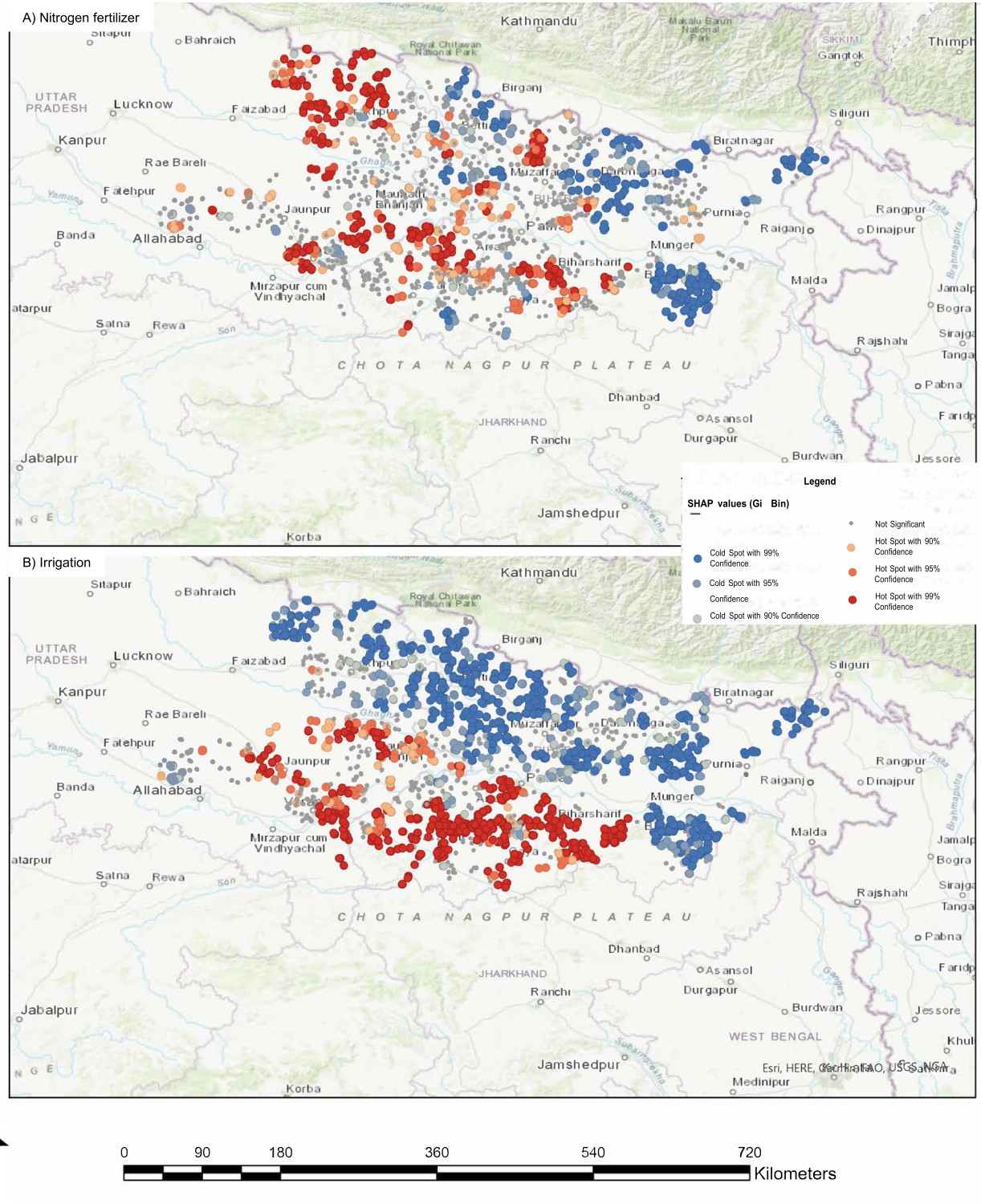

**Fig. 4 | Hotspot analysis of the two most important yield constraints in Eastern India.** Hot spot analysis of SHAP values for N fertilizer (**A**) and irrigation (**B**) in Eastern India. Locations mapped in blue have consistently negative SHAP values across fields, suggesting opportunities to intensify through improved management of the respective factors. Locations mapped in dark red have positive SHAP values, suggesting little scope to narrow yield gaps through changes in the respective management factors. Areas mapped in grey do not exhibit consistent responses across farm fields within a 10 km radius. Source data are provided as a Source Data file.

Scenario 2 demonstrates the power of an approach that 'learns' from cultivated landscapes to derive an analytics-based blanket N rate rather than extrapolating recommendations from experimental stations (i.e., Scenario 1). Nevertheless, achieving the production gains predicted in Scenario 2 would require a large total investment in N but is only anticipated to generate incremental increases in average yield and profitability for farmers adopting new practices. In contrast, the power of solution targeting is evident in Scenario 3 where around half the fields adopt new N rates but with a doubling of mean yield and profitability gains for the fields implementing new practices above the

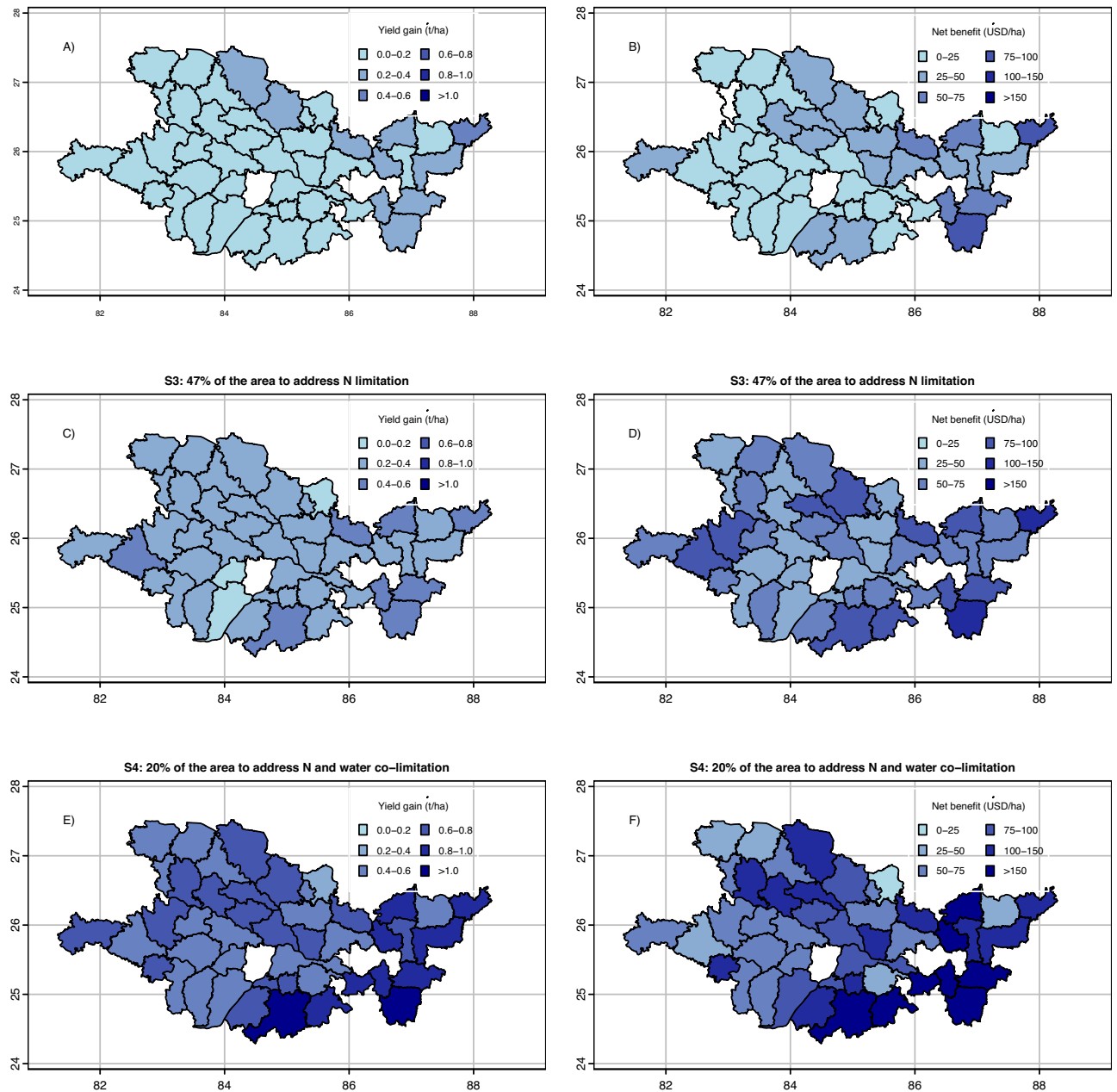

**Fig. 5 | Ex-ante scenario analysis associated with practice change and targeting strategies in Eastern India.** District average yield (t ha⁻¹) and profitability (USD ha⁻¹) gains for fields adopting new practices under Scenario 2 (**A**, **B**), Scenario 3 (**C**, **D**), and Scenario 4 (**E**, **F**). Scenario 2 consists of the analytics-informed blanket approach where all fields applied 180 kg N ha⁻¹, whereas Scenarios 3 and 4 use targeting criteria based on anticipated responsiveness to management changes. Scenario 3 increased the N rate only for fields with negative SHAP values for N, i.e., in I⁻N⁻ and I⁺N⁻ clusters. Scenario 4 addresses the co-limitation of N and irrigation for fields in the I⁻N⁻ cluster, i.e., increasing the N rate to 180 kg N ha⁻¹ and number of irrigations to 5. Source data are provided as a Source Data file.

gains in Scenario 2. The power of solution targeting is even more impactful when addressing multiple production constraints. Predictions from Scenario 4 suggest that if nitrogen and irrigation co-limitations are addressed in Eastern India, rice yield and profitability gains will triple (20% of total area) for fields implementing new practices.

Even though spatially explicit targeting in Scenarios 3 and 4 would not result in more total rice production than the blanket recommendation in Scenario 2, it could prove essential for sustainable intensification for three reasons. First, benefits from change need to be tangible for farmers and not expose them to higher levels of risk[42–44]. Second, even with the emergence of pluralistic extension systems and digital tools, many smallholders are not adequately connected to formal sources of knowledge[11]. With opportunities for productivity gains more clearly identified, scarce resources to support agricultural innovation can be focused where the returns on investment will likely be substantial. Lastly, targeting will also improve input use efficiencies. N use efficiencies in Scenarios 3 and 4 were significantly higher than with blanket recommendations, implying that a targeted approach will ensure that intensification does not undermine environmental sustainability goals, including greenhouse gas mitigation[45].

It is important to note that the yield models developed in this study do not fully capture the observed variance in rice yield, implying that there is more to learn when devising intensification pathways for rice cropping systems in India. Accordingly, the further development of efficient methods for characterizing cropping systems and environmental factors at scale are essential to support sustainable intensification within and beyond India. Moreover, the potential role of emerging technologies is not captured in our analytics-based approach, hence on-farm research networks must remain a vital component of assessing the performance of new technologies across crop production contexts. The on-farm validation of targeted recommendation and incorporating on-farm feedback back to analytics can make the approach more robust.

Finally, to translate analytics-led targeting strategies into practical recommendations will require simplified 'rules of thumb' to guide action for circumstances where full characterization data for individual fields is lacking. For example, rice fields with less than 4 irrigations and fertilizer N rates less than 118 kg N ha$^{-1}$ composed almost all of the fields in our case study region targeted for practice change in Scenario 4 where both irrigation and N are predicted to limit yields (Supplementary Fig. 4). Simplified recommendations in tandem with efforts to recognize and address the bottlenecks that farmers face when implementing new practices (see Urfels et al.[46,47]) will help accelerate efforts to narrow yield gaps in production contexts where intensification is needed to address sustainable development goals.

## Methods

The research conducted herein was reviewed by and complies with standards established by the Research Ethics Committee of the International Maize and Wheat Improvement Center (CIMMYT) as described in policy number DDG-POL-04–2019. The ethics review code for this study is IREC.2019.06. Verbal consent was obtained from all survey participants.

### Landscape-scale crop assessment surveys

The study area comprises the seven major rice producing states of India, namely Eastern Uttar Pradesh and Bihar ($n = 10,714$ field-year combinations), Odisha ($n = 747$), Jharkhand ($n = 717$), Chhattisgarh ($n = 1099$), West Bengal ($n = 1363$), and Andhra Pradesh ($n = 1046$) during the 2017, 2018, and 2019 monsoon seasons (Supplementary Fig. 5). Data analyses consisted of the following steps. First, attainable yield gaps (Yg$_a$) were estimated for each state. Second, random forest analytics were developed to identify the most important variables explaining yield outcomes in each state. Third, the SHapely Additive exPlanation (SHAP) was used to segregate the relative contribution of each production factor to rice yield prediction and, finally, machine learning-based scenario analysis was used to quantify the benefits of integrated crop management practices at regional level, with a focus in Bihar and adjacent areas of Eastern Uttar Pradesh where data collection was most intensive. Results of this analysis were further placed in a spatial context for sustainable rice intensification in the region through a geographical hotspot analysis.

The landscape-scale crop assessment surveys were conducted with digital collection tools and requested information on agronomic management practices and biophysical characteristics for the largest rice production field in each farm. The farmer reported yield was verified by measured crop cut yield through harvesting a $2 \times 2$ m quadrant randomly from the representative center of the field from a fraction of farms. Survey data and the corresponding data collection tool are freely available online[12]. The details of the sampling and data collection protocols are reported elsewhere[12]. Survey data for each field was then combined with gridded daily weather data from the reported sowing to harvest dates from NASA Power (https://power.larc.nasa.gov). Descriptive statistics of the variables used in the analysis are presented in Supplementary Table 2.

### Yield gap diagnostics at state level

The attainable yield gap (Yg$_a$) was estimated as the difference between the mean actual yield across highest yielding fields (i.e., top 10 percentile of the yield distribution; the attainable yield) in each state and the actual yield observed in all other fields in the respective state. Thereafter, state-specific random forest models were developed, and fine-tuned following Nayak et al.[14], to identify the most important factors explaining rice yield variability in each state. Model over-fitting was avoided by keeping at least 50 observations in all terminal nodes within each tree.

After each model was built, individual conditional expectance (ICE) plots[48] were created for the two most important management practices, sequentially, as identified by permutation-based feature importance. ICE plots were developed for individual fields to predict the relationship between the most important input variables and rice yield, while keeping all other input variables at their reported value for each field. For instance, suppose N fertilizer rate was identified as the most important variable to explain rice yield variability, then crop yield was predicted with ICE for a vector of N application rates capturing the range of N application rates observed in the data in steps of 10 kg N ha$^{-1}$. The difference between the predicted yield at the reported N application rate and the maximum yield across the vector of N application rates (Y$_{step1}$) reflects the expected yield gap closure once the most important constraint is addressed (Yg$_1$). Subsequently, the original feature values (i.e., N application rate, in this example) for each field were replaced with the corresponding N application rate associated with the maximum yield from the ICE (N$_{yld\_max}$) and ICE plots were created again for the second most important variable in combination with N$_{yld\_max}$ and the reported values of all other variables used in the model for each field (Y$_{step2}$). The difference between the Y$_{step2}$ and Y$_{step1}$ is defined as the expected yield gap closure once the second most important constraint is addressed (Yg$_2$). The combination of Yg$_1$ and Yg$_2$ indicates the maximum expected yield gap closure after removing yield constraints associated with the two most important variables explaining yield variability.

The distribution of Yg$_1$ and Yg$_2$, and their sum, was expressed relative to the attainable yield estimated for each state to establish the share of Yg$_a$ accounted by the two most important management practices. The yield gap analysis was conducted with the ranger and caret R packages[49,50] and with the iml R package[51].

### Eastern India region case study

A SHapley Additive exPlanation (SHAP)-based methodology was further deployed to quantify the relative contribution of biophysical factors and management practices to rice yields prediction for 10,714 field-year combinations in Bihar and adjacent districts in Eastern Uttar Pradesh ('Eastern India'). SHAP is a post-hoc methodology to interpret random forest models and identify heterogeneous effect of management practices on yield prediction. SHAP is based on cooperative game theory and is used to estimate the marginal contribution of each player to a team's overall performance[52]. By conceptualizing individual fields as a 'team' and management practices as 'players', the SHAP methodology can be used to quantify the relative contribution of each management practice to crop yield outcomes on individual field. Hence, with this methodology, the contextual value of different management practices can be translated into pathways for increasing crop productivity through condition-specific interventions.

SHAP is an additive feature attribution method that is used as a post-hoc approach for local interpretation of data-driven models. In short, it defines the contribution of individual variables to model predictions of the outcome of interest[52]. Thus, the SHAP value for any variable $J$ ($\phi_j$; t ha$^{-1}$) can be interpreted in our assessment as the marginal contribution of variable $J$ on rice yield prediction, as compared to the average predicted rice yield across the dataset. In other words, SHAP values refer to the yield contribution of individual variables,

expressed as either a positive or negative deviation from the population mean. Variables with larger positive and negative SHAP values in absolute terms have a large positive and negative influence on modeled yield predictions, respectively. The SHAP values were obtained for each field with the iml R package[51] and visualized in relation to the scaled absolute variable values. Numeric variables were normalized using minimum-maximum scaling, and categorical variables were ordinally factored and scaled (Supplementary Table 1). The spatial distribution of the two management practices with highest absolute SHAP value in Bihar and eastern Uttar Pradesh was assessed through a hot spot analysis conducted in ArcGIS Pro 2.9.0 and consisting of the calculation of the Getis-Ord Gi* statistic for each field assuming a fixed distance band of 10 km.

Absolute SHAP values for each input variable were averaged across fields to rank variables from most to least important at state level. Fields were further segregated into four clusters based on the two most important management practices at state level to delineate where a single versus multiple production practice changes are important to increase rice productivity. The clustering analysis supports a targeted approach to sustainable intensification, as opposed to a blanket approach where all fields receive the same intervention. The clustering was done based on the SHAP value for number of irrigations and N fertilizer rate for each field. The clusters included fields with a positive SHAP value for both practices ($I^+N^+$), a negative SHAP value for both practices ($I^-N^-$), and a positive SHAP value for one and a negative SHAP value for the other practice ($I^+N^-$ and $I^-N^+$). As such, positive SHAP values for a given management practice can be considered less yield limiting and a negative SHAP values more yield limiting, indicating where improved management can generate yield gains.

Four sustainable intensification scenarios were designed to explore the aggregated production benefits, additional input requirements, and profitability of yield gap closure in Bihar and Eastern Uttar Pradesh as compared to current farmers' practice. Scenario 1 consisted of the state level blanket recommendation of 125 kg N ha⁻¹ in all fields. Scenario 2 consists of blanket use of 180 kg N ha⁻¹, defined based on the partial dependency plots of a random forest model fitted to the pooled data for Bihar and Eastern Uttar Pradesh. There were two cluster-based targeting scenarios. Scenario 3 considered interventions for the N fertilizer only in the $I^+N^-$ and $I^-N^-$ clusters. The same N dose of 180 kg N ha⁻¹ was used in these clusters, whereas N rates were not changed in the other clusters. Scenario 4 consisted of cluster-based targeting for both N and irrigation, i.e., addressing the co-limitation of both production factors in a limited number of farms. The N use and irrigation in $I^-N^-$ cluster was changed to 180 kg N ha⁻¹ and 5 irrigations. Crop management remained unchanged in the other clusters. For Scenarios 1 and 2, the aggregated benefit in terms of additional production was obtained by multiplying the predicted yield increases (t ha⁻¹) by total rice area of each district. For Scenarios 3 and 4, the additional rice yield, additional water and N use, and returns on investment (i.e., additional resource costs, $20 USD per irrigation and $0.14 USD per kg subsidized N, subtracted from additional rice sales revenues based on the rice minimum support price for 2018) were estimated at the district level considering the share of farms in each cluster where interventions were targeted.

### Software

All data analysis were conducted in R (4.2.3) with the following package and version number, dplyr (1.1.4), caret (6.0.93), range (0.14.1), iml (0.11.1), geodata (0.5.3), terra (1.7.55), tidyverse (1.3.2), ggpubr (0.6.0) and dependencies, data.table (1.14.2), and gridExtra (2.3). Hotspot analysis was carried out in ArcGIS Pro v.2.9.0.

### Reporting summary

Further information on research design is available in the Nature Portfolio Reporting Summary linked to this article.

### Data availability

The data used in the analyses presented in the manuscript are provided in Supplementary Data 1-2. The boundary map can be downloaded from the geodata package, and the open street map used with ArcGIS can also be obtained from ArcGIS. The rice area obtained from https://aps.dac.gov.in is also provided in supplementary files. Source data are provided with this paper.

### Code availability

The R script used for the analysis is attached along with the manuscript in Supplementary Code 1.

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

## Acknowledgements

This work was made possible by long-term funding from the Bill and Melinda Gates Foundation (BMGF grant numbers OPP1052535 and OPP1133205; A.J.M., P.C.) and the US Agency for International Development (USAID grant number BFS-G-11-00002; A.J.M.) to the Cereal Systems Initiative for South Asia (https://csisa.org/). Additional support was provided by the One CGIAR Initiative on Excellence in Agronomy (INV-005431; K.T., J.V.S., A.J.M.). Accordingly, we would like to thank funders supporting research through contributions to the CGIAR Trust Fund: https://www.cgiar.org/funders/. This study builds upon the generous participation in surveys from thousands of smallholders in India, together with the diligent efforts of field scientists from ICAR's Krishi Vigyan Kendra (KVK) system and the CSISA team. Any opinions, findings, conclusions, or recommendations expressed in this publication are those of the authors and do not necessarily reflect the views of BMGF, USAID, or the CGIAR.

## Author contributions

H.S.N., A.J.M., J.V.S., and V.K. designed and implemented the study, conducted analyses, and drafted the first version of the manuscript. P.C., S.K.D., A.K.N., C.M.P., P.P., S.P., K.T., R.K.M., A.U. and U.S.G. reviewed and revised the manuscript, including the discussion and interpretation of the results. A.J.M., P.C., K.T. and J.V.S. mobilized funding to support the study.

## Competing interests

The authors declare no competing interests.
