## [Peer Review File · Nature Communications]

Context-dependent agricultural intensification pathways to increase rice production in IndiaREVIEWER COMMENTS

Reviewer #1 (Remarks to the Author):

Review of NCOMMS-24-12-12114

This study uses a large database collected from farmer rice fields in India to identify causes for yield gaps and opportunities to improve input-use efficiency via better management of nitrogen fertilizer and irrigation water. The authors show that there is room to increase current farmer yields by 50% through better agronomic practices and opportunities to be more efficiency in the use of water resources and nitrogen fertilizer, but the latter need to be tailored to each field in order to be effective at increasing yield and/or efficiency. The methodology, based on ML techniques, is robust and findings support the conclusions and claims of the study.

My major concern about this study is the novelty. Whereas the extent of the database and number of observations is impressive, how this study adds value to the established literature is not clear.

First, the analysis of farmer data as an approach to identify opportunity for yield increase is already established, for India and elsewhere (some examples: Jain et al., 2017; Tseng et al., 2021, Devkota et al 2021; McDonald et al 2022; Nayak et al 2022a, 2023), including studies ML techniques, so there is nothing new in the methodology by itself.

Second, the large yield gaps for rice in India, the opportunities to optimize associated water and nutrient management, and the importance of context-dependent interventions have already been documented for India, including publications by some of the authors of this paper (some examples: Aggarwal et al., 2008; Stuart et al., 2016; Yuan et al 2022; Nayak et al 2022b; Jha et al., 2018, 2022; Gerber et al., 2024).

Hence, the novelty of the article is marginal at best – this, together with the way the article is framed and written, probably makes it more appropriate for a discipline-oriented journal such as Field Crops Research or Agricultural Systems.

Other comments:

The definition of attainable yield seems arbitrary and theoretically debatable. Using the average yield of the top 10% highest-yielding fields seems arbitrary (why not 5% or 20%?) and, perhaps more importantly, how do we know that these farmers can be taken as a reference point for the rest of the population of farmers?

I could not find any data/code availability statement in the article – it would be useful to have access to them so that reviewers and eventually readers of the article can corroborate the findings by themselves. Coming from a major international research center, I want to believe that the database is publicly available somewhere but, again, I could not find this information in the

manuscript.

Reviewer #2 (Remarks to the Author):

General comments

This paper is interesting and well written. It seeks to provide policy-relevant insight to prioritize R&D investments in agriculture in East India through identification of factors with greatest influence on rice yield gaps. Rice is the primary staple food crop for 250 million people in that region. The authors state on lines 71-72: “In this study, we illustrate how big data and machine learning approaches (hereafter also referred to as ‘analytics-based’ methods) can substantially enhance the targeting of agricultural investments.” The goal of producing policy-relevant research results is laudable, and the paper produces some policy-relevant findings, but as currently written the paper is not ready for publication without major revisions that address the following concerns. The good news is that it should be possible to adequately address these issues in a revised paper.

First, while the authors use a big data/machine learning/analytics-based approach to identify the causes of yield gaps, there is nothing new in doing so. Other methods using similar approaches have been developed for this purpose with good success, including a substantial body of previously published research using large datasets with similar farmer survey data, is completely ignored. Likewise, much of the Discussion section extols the virtues of the “context-specific management approach” to technology transfer as if it is a brand-new opportunity. However, failing to cite previous work using similar methods and approaches, and especially work targeting the improvement of rice yields through context-specific management, weakens the case for publishing this paper. A list of example publications is given below. If the authors believe their approach is novel in some way and wish to comment on its novelty, they must acknowledge some of this previous work and explain how the methods used in their paper compare with those used in previous studies.

Second, the paper attempts to focus the reader’s attention on “intensification” as given in the title and at numerous locations in the text (see examples below under “Specific comments and concerns”). Sometimes it is referred to as “sustainable intensification”. But the authors do not define what intensification is and careful reading of the text shows that the authors simply mean achieving higher yields through closure of yield gaps. The title should be changed to more accurately reflect the purpose of the paper which is to improve rice production in East India by closing yield gaps using technologies already available to rice farmers. In the revised paper, mention of intensification, or sustainable intensification, should be avoided because that is not what this work is evaluating. Instead, this research estimates the explanatory power of production factors like N, P, and K fertilizer inputs, or number of irrigations in explaining the observed variation in farmer yields, and from that, the estimated increase in yields (i.e. yield gap closure) that would come from improving those practices.

Third, there is no mention of environmental impacts or economic benefits from widespread

adoption of the yield improving factors they identify, and especially the need for increased N fertilizer application and more irrigation to close yield gaps. For example, even in the best-performing scenario evaluated in this study (scenario 4), N fertilizer efficiency is very low and likely to result in a large portion of the additional applied N lost as nitrous oxide, a powerful GHG. See specific comments about this issue below. The text in Materials and Methods states, “For Scenarios 3 and 4, the additional rice yield, additional water and N use, and the additional returns on investment (additional resource use multiplied with resource cost (~20 USD per irrigation and 0.14 USD per kg N) subtracted from additional rice yield produced multiplied by rice minimum support price during 2018) to the different interventions were estimated at district level considering the share of farms in each cluster where interventions were considered.” Seems to me the prices quoted for N fertilizer are highly subsidized, which in large part is why use of N fertilizer at such low efficiency levels gives an economic return in the scenario 3 and 4 analyses. If N fertilizer prices are subsidized, the degree of subsidization should be stated.

Fourth, three supplementary figures and one supplementary table were included in the main text of the paper I reviewed, but I assume these supplementary materials are not intended to be included in the main text of the published paper. This is highly unusual although the supplemental figures and table are useful. Supplementary figure 3 is particularly informative and of interest. Perhaps it can be included in the main text and not in the supplemental information section?

Specific comments and concerns:

1. Inappropriate emphasis on the “analytics-based solution” approach used in this paper because as noted under General comment #1 above, there have been a number of other “analytics-based solutions” published using farm data (see list of citations below). Indeed, “analytics-based solution” is far too general to be worthwhile given the existence of other methods that can also be considered “analytic-based solutions”.

Lines 88-92: “The primary objectives of this study were two-fold: (i) to quantify the nature and causes of attainable rice yield gaps across seven major rice producing states of India, and (ii) to assess how analytics-based solution targeting may contribute to sustainable intensification through a case study of Bihar State and adjacent districts of Uttar Pradesh (hereafter referred to as ‘Eastern India’).”

Examples of studies that used farmer-reported data to identify yield limiting factor and their spatial delineation across scales from field to globe.

Andrade et al. 2022. *Agric. Sys.* [http://refhub.elsevier.com/S0378-4290\(23\)00135-1/sbref3](http://refhub.elsevier.com/S0378-4290(23)00135-1/sbref3)

Monzon et al. 2023. *Agric. Sys.* Agronomy explains large yield gaps in smallholder oil palm fields <https://doi.org/10.1016/j.agry.2023.103689>

Ratalino Edreira et al. 2020. *Field Crops Res.* [http://refhub.elsevier.com/S0378-4290\(23\)00135-1/sbref43](http://refhub.elsevier.com/S0378-4290(23)00135-1/sbref43)

Rizzo et al. 2023. *Field Crops Res.* A farmer data-driven approach for prioritization of agricultural research and development: A case study for intensive crop systems in the humid tropics,

2. Intensification versus closing yield gaps. See quoted text below in which intensification is used. All of them conflate closing yield gaps with intensification or sustainable intensification. Sustainable intensification includes an increased productivity component as well as an

ecosystems services component. In fact, the paper is about increasing yields through use of inputs such as fertilizer and irrigation; the paper does not address tradeoffs with ecosystems services other than food provisioning.

Lines 59-61: “Beyond on-station and on-farm research trials, recent advances in yield gap analysis offer another set of 60 methodologies for quantifying the sustainable intensification potential of agroecosystems through observational data”

Lines 65-69: “but focused on population-level analysis without considering how intensification opportunities vary across fields and between sub-regions. Recent advances in machine learning allows the identification of field-specific yield 69 constraints and can support the development of site-specific insights and strategies for closing yield gaps through ex-ante scenario analysis.”

Lines 108-110: “The sizeable yield gaps documented in these data indicate considerable scope to intensify rice production from existing land in India based on currently available technologies and management practices.”

Lines 187-189: “For example, at the district level, between 28 - 42% of fields were not limited by N nor by irrigation (I+N+) indicating scope for intensification through irrigation and N management in each district (Supplementary Figure 2).”

Lines 198-199: “After constructing a yield model and characterizing the key drivers of yield outcomes, we then compared different strategies for achieving sustainable rice intensification through a scenario analysis.”

Lines 259-260: “The Government of India has focused on rice intensification in regions where yield gaps are perceived to be high (e.g., ‘Bringing the Green Revolution to Eastern India’)”

3. The reasons for different rice yields in different Indian states and regions are given as a statement of fact without any supporting data or citations. Please provide one or two references supporting the statement below. Otherwise, delete text providing the putative causes of yield differences.

Lines 102-105: “Average rice yield across the surveyed Indian states ranged between 3.3 t ha⁻¹ in Jharkhand and 5.5 t ha⁻¹ 102 in Andhra Pradesh (Figure 1a). Jharkhand also had the lowest attainable yield (5.1 t ha⁻¹ 103) and Andhra Pradesh the highest at 7.7 t ha⁻¹ , reflecting underlying differences in rice production environments related to water resources and solar radiation received during the growing season”

4. Only a small proportion of variance in yields is explained in Odisha compared to Andhra Pradesh (see statement below). An explanation is needed. Likewise, such low explanatory power makes identification of most influential factors much more uncertain than in AP, and the authors should acknowledge this greater uncertainty.

Lines 118: “Random Forest (RF) models were developed to identify yield constraints for each state. RF explained between 29% (Odisha) and 52% (Andhra Pradesh) of the overall yield variation.”

5. Why such low N fertilizer efficiency? Although better than blanket recommendations, Scenario 3 still has a very low N fertilizer use efficiency (3 kg grain/kg applied N, assuming rice grain has an N concentration of 1.1%). Under typical irrigated rice production, N fertilizer use efficiency NFUE typically ranges from 20-40 kg grain/kg applied N. Even scenario 4, the best-case scenario in this study, has a low NFUE of 7, which means more than 90% of the applied N is not taken up by the crop and a large portion is lost to the environment as nitrous oxide or sometimes ammonia.

Moreover, low NFUE also means relatively low return on investment in the additional N fertilizer. The revised paper should mention these considerations.

Lines 220-225: “By targeting fields where yields were co-limited by N and irrigation (Scenario 4), practice changes were only implemented in 20% of all rice fields. Among this sub-population, simultaneous changes to N and irrigation management were predicted to produce an additional 0.56 million tons of rice with a modest investment of 0.08 million tons of N in combination with an increase in irrigation that ranged from 1 to 4 events per field, a change that scales at the regional level to 2.33 million additional irrigations per season, approximately equivalent to an average of 17% of the water safely available for future use, which varies across the districts³⁰ (SI Table 2).”

6. The goal of “demonstrating the importance of...” borders on being promotional and is not a viable objective for research. Please revise accordingly.

Lines 279-283: “To demonstrate the importance of a targeted and analytics-based approach, we developed a machine learning model for rice yield to predict productivity outcomes under hypothetical scenarios of change for N and irrigation management in Bihar and Eastern Uttar Pradesh region of Eastern India, a case study region where these two factors were the top contributors to attainable yield gaps (see Figure 2) with a strong spatial dependence (Figure 4).”

Reviewer #3 (Remarks to the Author):

The authors have effectively applied the SHAP technique following the training of the Random Forest algorithm and have presented their SHAP summary plots. However, there is still scope for further consideration in this article. Since the SHAP technique provides valuable insights into the local interpretation of models, it is necessary to go beyond the summary SHAP plots. For example, using SHAP dependence plots, the authors may interpret the relationship between the sample values of any independent variable and their corresponding SHAP values to understand "critical thresholds" where the contribution of that independent variable changes from positive to negative or vice versa. Additionally, authors can utilise SHAP force plots to illustrate the contribution of variables for single selected samples. Essentially, the authors solely relied on SHAP summary plots to rank feature importance and did not extensively discuss how their input variables behave differently with varying values. Utilising other plots would allow for a more in-depth interpretation, facilitating a discussion on how their findings complement, confirm, or contradict other studies.

#Appendix 1: Reply to reviewer

(Please see MS_colored.docx for the highlighted changes in revision)

Reviewer #1

This study uses a large database collected from farmer rice fields in India to identify causes for yield gaps and opportunities to improve input-use efficiency via better management of nitrogen fertilizer and irrigation water. The authors show that there is room to increase current farmer yields by 50% through better agronomic practices and opportunities to be more efficiency in the use of water resources and nitrogen fertilizer, but the latter need to be tailored to each field in order to be effective at increasing yield and/or efficiency. The methodology, based on ML techniques, is robust and findings support the conclusions and claims of the study.

Authors: We thank the reviewer for the positive appreciation of our manuscript, particularly that our methodology is robust and fit-for-purpose as also highlighted by Reviewer 3.

My major concern about this study is the novelty. Whereas the extent of the database and number of observations is impressive, how this study adds value to the established literature is not clear. First, the analysis of farmer data as an approach to identify opportunity for yield increase is already established, for India and elsewhere (some examples: Jain et al., 2017; Tseng et al., 2021, Devkota et al 2021; McDonald et al 2022; Nayak et al 2022a, 2023), including studies ML techniques, so there is nothing new in the methodology by itself.

Authors: We agree with the reviewer that there are several studies using farmer field data and machine learning techniques in agronomy. However, the studies cited by the reviewer focused on descriptive analytics to generate population level insights. The approach developed in the current manuscript is a significant departure from those studies. It combines field-specific insights, solution targeting for single and multiple production constraints, and predictive analytics to assess the value of different rice development strategies in a spatial framework across multiple dimensions (yield, profitability, NUE). The proposed method is therefore new and adds value to existing studies on yield gap analysis using farmer field data. More specifically, our work offers methodological advances and provides novel insights in the following ways:

*(a) this is the first comprehensive assessment of sub-national rice yields in India where data-driven insights identify drivers of yield gaps in different production contexts (**Figures 2 and 3**). We note this is a very timely issue given the recent policies of the Government of India to temporarily ban rice exports over concerns about the adequacy of domestic supplies.*

*(b) this is the first detailed assessment of yield constraints in Eastern India at farm level (**Figure 3**), i.e., including diagnostic of field-specific insights and exploration of alternative futures through scenario analysis. The focus on Eastern India is justified since it is a priority region to the Indian government due to high dependency on rice and wheat farming and the largest concentration of rural poverty on a global basis.*

*(c) this is the first study using the SHAP methodology in agronomy, certainly so in the context of yield gap analysis with farmer field data. This allowed us to identify yield constraints for individual fields (**Figure 3 and 4**).*

(d) we used machine learning for ex-ante predictive analytics to evaluate yield and efficiency gains (nitrogen use efficiency and returns on investment) achievable with agronomic management strategies linked to different production contexts based on SHAP. As pointed by Reviewer 2, these results have strong policy relevance, especially since they are presented in a spatial framework to guide investment.

Given the above, we want to clarify that none of the studies mentioned by reviewer, or any other published study to our knowledge, overlaps with our analysis in terms insights for rice in India or our unique methodological approach. We further highlight these aspects in the revised manuscript to make them clearer to reader, see **Lines 65-68, L70-72, L82-84, L93-97, L100-101** in the colored version of MS.

Second, the large yield gaps for rice in India, the opportunities to optimize associated water and nutrient management, and the importance of context-dependent interventions have already been documented for India, including publications by some of the authors of this paper (some examples: Aggarwal et al., 2008; Stuart et al., 2016; Yuan et al 2022; Nayak et al 2022b; Jha et al., 2018, 2022; Gerber et al., 2024).

Authors: We thank the reviewer for this remark. The central role of water and nutrient management for rice yield gap closure is universally appreciated, as pinpointed by the reviewer. However, our work significantly advances the conversation by i) characterizing the intensity of constraints across a broad range of production conditions, ii) considering geographic distributions and other targeting criteria, and iii) assessing the implications of targeted and blanket approaches to water and nitrogen management to yield gap closure (see **Figure 4 and Lines 100-101**). We feel that the reviewer overlooked these critical aspects of our study and have made this clearer **in Lines 65-68, L70-72, L82-84, L95-97**.

We also reviewed all the literature noted by the reviewer and considered how our analysis differs from those earlier analysis:

1. Yuan et al., 2022. This study reports an analysis at the agro-ecological zone level for rice in SE Asia. The study does not decompose yield gaps into constituent factors and does not include India.
2. Jha et al., 2018, 2022. These studies focus on rice in Eastern India, but the methodology relies on stochastic frontier analysis at the population level. No ex-ante assessment is provided regarding the impact of addressing yield constraints on food security, nutrient use efficiency, or profitability.
3. Gerber et al., 2024. This study provides a global analysis on yield gap trends, hence no insight on yield constraints and associated yield gains at the farm level. The study provides no context for India either and no actionable interventions or ex-ante assessment associated with addressing yield constraints.
4. Nayak et al., 2022b. This study uses stochastic frontier analysis to decompose yield gaps in Northwest India (Punjab and Haryana) referring to the high productivity tracts of the Indo-Gangetic Plains, which benefited most from the Green Revolution. The study provides a population level analysis with no spatial component or ex ante assessment of different agronomic management strategies.
5. Stuart et al., 2016. This is a review article for rice yield gaps highlighting the methods used to estimate and explain yield gaps. No primary data or analysis is provided for India.

Hence, the novelty of the article is marginal at best – this, together with the way the article is framed and written, probably makes it more appropriate for a discipline-oriented journal such as Field Crops Research or Agricultural Systems.

Authors: The novelty aspect of our manuscript, as also highlighted in our previous responses, includes the following aspects:

1. *This is the first assessment of yield constraints for rice across seven Indian states, out of which six are among the top ten rice producing states in India (https://ipad.fas.usda.gov/cropexplorer/cropview/comm_chartview.aspx?fattributeid=1&cropid=0422110&startrow=1&sel_year=2022&ftypeid=47®ionid=sasia&cntryid=IND&nationalGraph=False). No other study used such data-driven approach to characterize sub-national yield gaps and yield gap drivers for rice in India.*
2. *This is the first comprehensive assessment of yield constraints in Eastern India, a priority region to the Indian government due to high dependency on rice and wheat farming and the largest density of population living below poverty line in the world. Please note that India is a big country and because there are studies in the states of Punjab and Haryana it doesn't mean such studies are relevant to Bihar, where farming systems and production constraints are different. Moreover, our analysis has direct policy relevance by placing results in a spatial framework, simultaneously addressing multiple production constraints, and evaluating management options across three sustainability dimensions. This policy relevance of our study was also highlighted by Reviewer 2.*
3. *To our knowledge, this is the first study using the SHAP methodology in agronomy certainly so for yield gap analysis at the scale of individual fields.*
4. *Beyond estimating SHAP values with random forest, we also conducted ex-post analyses on those estimates to cluster fields based on their extent of water and nutrient co-limitation. To our knowledge, no other study has done this to date.*
5. *Our analysis has a spatial dimension allowing us to unravel which yield constraints are prevalent where and to conduct an ex-ante quantification of aggregated benefits from practice change in a spatially explicit manner. Overall, these steps provide insights into which interventions should be prioritize where, and what are associated returns-on-investment of doing so, towards increasing rice productivity in Eastern India.*

*We clarify the novelty of the study in **Lines 65-68, L70-72, L82-84, L93-97, L100-101** and trust the points enumerated above clarify the novelty of our study in an unequivocal way and justify why we feel the readership of Nature Communications is the right audience for our manuscript.*

Other comments:

The definition of attainable yield seems arbitrary and theoretically debatable. Using the average yield of the top 10% highest-yielding fields seems arbitrary (why not 5% or 20%?) and, perhaps more importantly, how do we know that these farmers can be taken as a reference point for the rest of the population of farmers?

Authors: We thank the reviewer for this comment. We would like to clarify that none of our data-driven analysis used the attainable yield gap (top 10% farmers' yield minus actual yield) as dependent variable in the fitted models, hence this calculation has no implications for our results beyond the state-level yield gap estimation presented in Figure 1. As for the validity of using the 10% as a benchmark, that is a 'known unknown' for all yield gaps studies that either use biophysical potential or top-performing fields as high-end yield estimates of productivity potential. Nevertheless, by using machine learning-based scenario analysis, we do explicitly and quantitatively assess the relevance of specific practice changes to closing yield gaps and do not assume that a specific practice change will have commensurate benefits across all fields.

I could not find any data/code availability statement in the article – it would be useful to have access to them so that reviewers and eventually readers of the article can corroborate the findings by themselves. Coming from a major international research center, I want to believe that the database is publicly available somewhere but, again, I could not find this information in the manuscript.

Authors: Yes, data and code will be made available upon publication of the study according to journal guidelines.

Reviewer 2

General comments

This paper is interesting and well written. It seeks to provide policy-relevant insight to prioritize R&D investments in agriculture in East India through identification of factors with greatest influence on rice yield gaps. Rice is the primary staple food crop for 250 million people in that region. The authors state on lines 71-72: "In this study, we illustrate how big data and machine learning approaches (hereafter also referred to as 'analytics-based' methods) can substantially enhance the targeting of agricultural investments." The goal of producing policy-relevant research results is laudable, and the paper produces some policy-relevant findings, but as currently written the paper is not ready for publication without major revisions that address the following concerns. The good news is that it should be possible to adequately address these issues in a revised paper.

Authors: We thank the reviewer for the positive comments on our manuscript, and for highlighting that our analysis produced policy-relevant findings. We also thank the reviewer for the valuable suggestions that helped improve our manuscript, as explained below.

First, while the authors use a big data/machine learning/analytics-based approach to identify the causes of yield gaps, there is nothing new in doing so. Other methods using similar approaches have been developed for this purpose with good success, including a substantial body of previously published research using large datasets with similar farmer survey data, is completely ignored. Likewise, much of the Discussion section extols the virtues of the "context-specific management approach" to technology transfer as if it is a brand-new opportunity. However, failing to cite previous work using similar methods and approaches, and especially work targeting the improvement of rice yields through context-specific management, weakens the case for publishing this paper. A list of example publications is given below. If the authors believe their

approach is novel in some way and wish to comment on its novelty, they must acknowledge some of this previous work and explain how the methods used in their paper compare with those used in previous studies.

Examples of studies that used farmer-reported data to identify yield limiting factor and their spatial delineation across scales from field to globe.

Andrade et al. 2022. Agric. Sys. [http://refhub.elsevier.com/S0378-4290\(23\)00135-1/sbref3](http://refhub.elsevier.com/S0378-4290(23)00135-1/sbref3)
Monzon et al. 2023. Agric. Sys. Agronomy explains large yield gaps in smallholder oil palm fields <https://doi.org/10.1016/j.agry.2023.103689>

Ratalino Edreira et al. 2020. Field Crops Res. [http://refhub.elsevier.com/S0378-4290\(23\)00135-1/sbref43](http://refhub.elsevier.com/S0378-4290(23)00135-1/sbref43)

Rizzo et al. 2023. Field Crops Res. A farmer data-driven approach for prioritization of agricultural research and development: A case study for intensive crop systems in the humid tropics,

Authors: We thank the reviewer for providing this valuable feedback, which we also address in our response to reviewer #1.

The approach followed in the present study is a departure from the cited articles, but we agree that some can be sensibly acknowledged as background literature. Most of the mentioned studies rely on a global model, such as a single regression or conditional inference tree (e.g., Rizzo et al., 2023). Others do not provide diagnostics of yield constraints for individual fields or lack an ex-ante predictive assessment of practice change (e.g., Monzon et al., 2023).

*Our state-of-the-art method of local interpretation of a global model with SHAP provides a powerful next step for guiding policies and agricultural development programming, and the importance of this approach is reflected in the results of our ex-ante scenario analysis that draw on targeting with SHAP. We have reinforced these points in **Lines 261-264** of the manuscript and included the suggested citation as background information.*

As pointed out to Reviewer 1, the novelty of the study can be summarized as follows:

*(a) this is the first comprehensive assessment of sub-national rice yields in India where data-driven insights identify drivers of yield gaps in different production contexts (**Figures 2 and 3**). We note this is a very timely issue given the recent policies of the Government of India to temporarily ban rice exports over concerns about the adequacy of domestic supplies.*

*(b) this is the first detailed assessment of yield constraints in Eastern India at farm level (**Figure 3**), i.e., including diagnostic of field-specific insights and exploration of alternative futures through scenario analysis. The focus on Eastern India is justified since it is a priority region to the Indian government due to high dependency on rice and wheat farming and the largest concentration of rural poverty on a global basis.*

*(c) this is the first study using the SHAP methodology in agronomy, certainly so in the context of yield gap analysis with farmer field data. This allowed us to identify yield constraints for individual fields (**Figure 3 and 4**).*

(d) we used machine learning for ex-ante predictive analytics to evaluate yield and efficiency gains (nitrogen use efficiency and returns on investment) achievable with agronomic management strategies linked to different production contexts based on SHAP. As pointed by Reviewer 2, these

results have strong policy relevance, especially since they are presented in a spatial framework to guide investment.

Thus, the whole workflow described in this study is a brand-new approach to diagnose yield constraints in farmers' fields and assess ex-ante the impact of practice change. This contrasts sharply with previous studies relying on a global model for contextualization of yield constraints across a population of farms. The workflow will be highly relevant for future yield gap diagnostic studies and studies aiming at exploring paths for sustainable intensification in a cost-effective way.

Second, the paper attempts to focus the reader's attention on "intensification" as given in the title and at numerous locations in the text (see examples below under "Specific comments and concerns"). Sometimes it is referred to as "sustainable intensification". But the authors do not define what intensification is and careful reading of the text shows that the authors simply mean achieving higher yields through closure of yield gaps. The title should be changed to more accurately reflect the purpose of the paper which is to improve rice production in East India by closing yield gaps using technologies already available to rice farmers. In the revised paper, mention of intensification, or sustainable intensification, should be avoided because that is not what this work is evaluating. Instead, this research estimates the explanatory power of production factors like N, P, and K fertilizer inputs, or number of irrigations in explaining the observed variation in farmer yields, and from that, the estimated increase in yields (i.e. yield gap closure) that would come from improving those practices.

Authors: We thank the reviewer for raising the issue of our use of the term 'intensification' throughout the manuscript. Indeed, our focus was on evaluating where specific agronomic practice changes will result in yield gains (i.e., production intensification – more grain per unit area). We have now substituted 'yield' for 'intensification' in most places in the manuscript to add clarity.

Through ex ante scenario analysis at field scale, we do assess how different strategies for agronomic management can result in divergent productivity, nitrogen use efficiency, and profitability outcomes. In doing so, we identify strategies for increasing yields that also have additional environmental and profitability benefits. Such approaches are commonly understood as 'sustainable intensification' and we prefer to retain the terminology in the context of presenting results from the scenario analysis.

*We defined what we mean with intensification in **L41** and what we mean by sustainable intensification in **L42-43** and now use the concepts consistently across the manuscript.*

With reference to the title, we do in fact use ex ante assessment methods to quantitatively assess practice changes and implications for yield, profitability, and NUE; that is, we are not simply describing current reality, but rather using predictive analytics to identify intensification pathways. As such, we prefer to keep the current title.

*As suggested, we deleted the intensification word from **L67, L93-94, L101, L111, L137, L194-195, L300-301** (Colored yellow), and kept sustainable intensification, where it is more fit for purpose.*

Third, there is no mention of environmental impacts or economic benefits from widespread adoption of the yield improving factors they identify, and especially the need for increased N fertilizer application and more irrigation to close yield gaps. For example, even in the best-performing scenario evaluated in this study (scenario 4), N fertilizer efficiency is very low and likely to result in a large portion of the additional applied N lost as nitrous oxide, a powerful GHG. See specific comments about this issue below. The text in Materials and Methods states, “For Scenarios 3 and 4, the additional rice yield, additional water and N use, and the additional returns on investment (additional resource use multiplied with resource cost (~20 USD per irrigation and 0.14 USD per kg N) subtracted from additional rice yield produced multiplied by rice minimum support price during 2018) to the different interventions were estimated at district level considering the share of farms in each cluster where interventions were considered.” Seems to me the prices quoted for N fertilizer are highly subsidized, which in large part is why use of N fertilizer at such low efficiency levels gives an economic return in the scenario 3 and 4 analyses. If N fertilizer prices are subsidized, the degree of subsidization should be stated.

*Authors: We thank the reviewer for this remark, and we confirm that indeed the price of N fertilizer is highly subsidized in India (i.e., 90% of the actual cost is government subsidized). This is now mentioned in **Lines 423-424**. We also agree that highly subsidized fertilizer prices can lead to reductions in NUE and increase the risk of associated GHG emissions. Yet, we would like to note that the NUE is not unusually low as suggested by the reviewer; please reference our response to comment number 5 below.*

*However, we respectfully disagree regarding the point of not mentioning environmental impacts and economic benefits associated with practice changes. This analysis was conducted using predictive analytics through scenario analysis to assess the impact of practice change on nitrogen-use efficiency and profitability associated with different approaches to agronomic management with results are presented in **Figure 5 and Supplementary Table 1**.*

Fourth, three supplementary figures and one supplementary table were included in the main text of the paper I reviewed, but I assume these supplementary materials are not intended to be included in the main text of the published paper. This is highly unusual although the supplemental figures and table are useful. Supplementary figure 3 is particularly informative and of interest. Perhaps it can be included in the main text and not in the supplemental information section?

Authors: Thank you for the suggestion. Yes, we moved the supplementary information to a separate file while preparing the final version of the manuscript. However, we prefer to keep SI Figure 3 as supplement to the manuscript, since the figure contains the aggregated results at district level from Figure 5, hence partly duplicating its results in the main manuscript.

Specific comments and concerns:

1. Inappropriate emphasis on the “analytics-based solution” approach used in this paper because as noted under General comment #1 above, there have been a number of other “analytics-based solutions” published using farm data (see list of citations below). Indeed, “analytics-based solution” is far too general to be worthwhile given the existence of other methods that can also be considered “analytic-based solutions”.

Lines 88-92: “The primary objectives of this study were two-fold: (i) to quantify the nature and causes of attainable rice yield gaps across seven major rice producing states of India, and (ii) to assess how analytics-based solution targeting may contribute to sustainable intensification through a case study of Bihar State and adjacent districts of Uttar Pradesh (hereafter referred to as ‘Eastern India’).”

*Authors: We are not claiming to have invented, or to represent the only, ‘analytics based’ approach to identifying agronomic solutions. We are simply using this terminology to describe our approach in a succinct way and, accordingly, have transparently defined the term in **Lines 73-74** to imply the combination of big data and machine learning.*

2. Intensification versus closing yield gaps. See quoted text below in which intensification is used. All of them conflate closing yield gaps with intensification or sustainable intensification. Sustainable intensification includes an increased productivity component as well as an ecosystems services component. In fact, the paper is about increasing yields through use of inputs such as fertilizer and irrigation; the paper does not address tradeoffs with ecosystems services other than food provisioning.

Authors: We thank the reviewer for this comment. We defined intensification as yield increase with changes in agronomic practices. Further we kept the word sustainable intensification where it is more fit for purpose. Please refer to a detailed explanation on the reply to second general comment for the modifications done in the manuscript to address these concerns. Please also see the below cases, where we changed the words.

Lines 59-61: “Beyond on-station and on-farm research trials, recent advances in yield gap analysis offer another set of 60 methodologies for quantifying the sustainable intensification potential of agroecosystems through observational data.”

Authors: We prefer to keep sustainable intensification here, as this is a general statement.

Lines 65-69: “but focused on population-level analysis without considering how intensification opportunities vary across fields and between sub-regions. Recent advances in machine learning allows the identification of field-specific yield 69 constraints and can support the development of site-specific insights and strategies for closing yield gaps through ex-ante scenario analysis.”

*Authors: We agree with the suggestion here. We changed intensification opportunities to how yield constraints vary. See **L67**.*

Lines 108-110: “The sizeable yield gaps documented in these data indicate considerable scope to intensify rice production from existing land in India based on currently available technologies and management practices.”

*Authors: We agree with the suggestion here. Here we used the word “increase” in place of “intensify”. Please see **L111**.*

Lines 187-189: “For example, at the district level, between 28 - 42% of fields were not limited by

N nor by irrigation (I+N+) indicating scope for intensification through irrigation and N management in each district (Supplementary Figure 2).”

Authors: This sentence is now deleted as a part of revision. Please see L194-195.

Lines 198-199: “After constructing a yield model and characterizing the key drivers of yield outcomes, we then compared different strategies for achieving sustainable rice intensification through a scenario analysis.”

Author: Here also we intent to keep the word sustainable intensification, as through predictive analytics and scenario analysis, we show how targeting can improve yield, nitrogen-use efficiency, and profitability for farmers adopting new practices. These are core tenants of sustainable intensification. Thus, we prefer to keep sustainable rice intensification in this sentence.

Further, as advised by the reviewer, we avoided intensification in most of the cases in the revised manuscript.

Lines 259-260: “The Government of India has focused on rice intensification in regions where yield gaps are perceived to be high (e.g., ‘Bringing the Green Revolution to Eastern India’)”

Authors: We do prefer to keep intensification here, as this is the terminology and focus of the Government of India.

3. The reasons for different rice yields in different Indian states and regions are given as a statement of fact without any supporting data or citations. Please provide one or two references supporting the statement below. Otherwise, delete text providing the putative causes of yield differences.

Lines 102-105: “Average rice yield across the surveyed Indian states ranged between 3.3 t ha⁻¹ in Jharkhand and 5.5 t ha⁻¹ in Andhra Pradesh (Figure 1a). Jharkhand also had the lowest attainable yield (5.1 t ha⁻¹) and Andhra Pradesh the highest at 7.7 t ha⁻¹, reflecting underlying differences in rice production environments related to water resources and solar radiation received during the growing season”.

Authors: We agree with the reviewer’s remark. We deleted the putative cause of the yield differences reported. Please see L107-108.

4. Only a small proportion of variance in yields is explained in Odisha compared to Andhra Pradesh (see statement below). An explanation is needed. Likewise, such low explanatory power makes identification of most influential factors much more uncertain than in AP, and the authors should acknowledge this greater uncertainty.

Lines 118: “Random Forest (RF) models were developed to identify yield constraints for each state. RF explained between 29% (Odisha) and 52% (Andhra Pradesh) of the overall yield variation.”

Authors: We agree with the reviewer’s remark. This was indeed acknowledged in the Discussion section. Please see Lines 306-307.

5. Why such low N fertilizer efficiency? Although better than blanket recommendations, Scenario 3 still has a very low N fertilizer use efficiency (3 kg grain/kg applied N, assuming rice grain has

an N concentration of 1.1%). Under typical irrigated rice production, N fertilizer use efficiency NFUE typically ranges from 20–40 kg grain/kg applied N. Even scenario 4, the best-case scenario in this study, has a low NFUE of 7, which means more than 90% of the applied N is not taken up by the crop and a large portion is lost to the environment as nitrous oxide or sometimes ammonia. Moreover, low NFUE also means relatively low return on investment in the additional N fertilizer. The revised paper should mention these considerations.

Lines 220-225: “By targeting fields where yields were co-limited by N and irrigation (Scenario 4), practice changes were only implemented in 20% of all rice fields. Among this sub-population, simultaneous changes to N and irrigation management were predicted to produce an additional 0.56 million tons of rice with a modest investment of 0.08 million tons of N in combination with an increase in irrigation that ranged from 1 to 4 events per field, a change that scales at the regional level to 2.33 million additional irrigations per season, approximately equivalent to an average of 17% of the water safely available for future use, which varies across the districts³⁰ (SI Table 2).”

*Authors: We thank the reviewer for this comment and for the opportunity to clarify the ambiguity in the NUE values that can be calculated from Supplementary Table 2. These values represent changes in yield and input use as they diverge from current farmer practices and are not total values. As such, the overall fertilizer NUE is much higher than the incremental value the reviewer is estimating from the table (i.e., 3 kg grain/kg applied N for Scenario 3). If full yield and N rate are accounted for Scenario 3, we estimate NUE at 22.7 kg grain kg⁻¹ N, a value within the reported range of 20-40 kg grain kg N⁻¹ noted by the reviewer. We explain this further in the **caption of SI Table 2**.*

Let's take an example to better explain how to interpret our results. The predicted yield in S3 was 3778 kg ha⁻¹ and the mean N rate was 87.6 kg N ha⁻¹. Scenario 3 assumed the N rate was increase to 180 kg N ha⁻¹, corresponding to an increase of ca. 92 kg N ha⁻¹, whereas yield increased to 4080 kg ha⁻¹, corresponding to 302 kg ha⁻¹ yield increase. The additional yield increase relative to N increase is then 3.28 kg additional yield kg⁻¹ additional N. But the partial factor productivity (which the reviewer refers to as NUE) is actually 4080 / 180 = 22.7 kg grain kg⁻¹ N applied, hence.

6. The goal of “demonstrating the importance of...” borders on being promotional and is not a viable objective for research. Please revise accordingly.

Lines 279-283: “To demonstrate the importance of a targeted and analytics-based approach, we developed a machine learning model for rice yield to predict productivity outcomes under hypothetical scenarios of change for N and irrigation management in Bihar and Eastern Uttar Pradesh region of Eastern India, a case study region where these two factors were the top contributors to attainable yield gaps (see Figure 2) with a strong spatial dependence (Figure 4).”

*Authors: We agree with the suggested changes and implemented them accordingly (see **Lines 278-279**). We adopted “to determine the potential of” instead of “to demonstrate the potential of”.*

Reviewer #3:

The authors have effectively applied the SHAP technique following the training of the Random Forest algorithm and have presented their SHAP summary plots. However, there is still scope for

further consideration in this article. Since the SHAP technique provides valuable insights into the local interpretation of models, it is necessary to go beyond the summary SHAP plots. For example, using SHAP dependence plots, the authors may interpret the relationship between the sample values of any independent variable and their corresponding SHAP values to understand "critical thresholds" where the contribution of that independent variable changes from positive to negative or vice versa. Additionally, authors can utilize SHAP force plots to illustrate the contribution of variables for single selected samples. Essentially, the authors solely relied on SHAP summary plots to rank feature importance and did not extensively discuss how their input variables behave differently with varying values. Utilizing other plots would allow for a more in-depth interpretation, facilitating a discussion on how their findings complement, confirm, or contradict other studies.

Authors: We thank the reviewer for the evaluation of the SHAP methodology as used in our manuscript. It is reassuring that we applied it effectively in our analysis and given the purpose of our study. We would like to note that we used the SHAP values for the following analysis:

- (a) SHAP summary plot to rank yield constraints (**Figure 3**). Please note the color code in the plot indicates the direction of each variable originally reported in the dataset (hence, we did show how the value of input variables relates to the SHAP values).*
- (b) SHAP values were used to cluster farms based on the level of colimitation of the two most important management variables (irrigation number and N applied) – see **Supplementary Figure 1 and 2 and L402-405**.*
- (c) SHAP values were also in a hotspot analysis to unravel how yield constraints across the region-of-interest (**Figure 4**).*
- (d) SHAP values were used in an ex-ante assessment of the impact of practice change on production outcomes, NUE, and return-on-investment (**Figure 5**).*
- (e) Based on the SHAP dependence plot we had derived the critical threshold, where the contribution of two most important management practices, irrigation number and N applied, changed from negative to positive and we had already mentioned those in **L315-317**.*

*The above five analysis all refer to figures and text in the main manuscript. We also investigated the SHAP plots for the two most important management variables, which was not included in the previous version of the manuscript and is now included in **SI Figure 3**. We now added those additional figures to the SI with the relationship between number of irrigations and N applied and the respective SHAP values, which was already discussed in the manuscript (see **Lines 315-317**). Having said this, we agree there are many other uses of the methodology, but we feel these were the most relevant ones given the purpose of our manuscript.*

REVIEWERS' COMMENTS

Reviewer #2 (Remarks to the Author):

The authors should be commended for greatly improving their paper in a number of ways through revisions made in response to the three external reviews. I recommend publication pending minor, but important, revision. I would not recommend publication without making these revisions. I hope they make the appropriate revisions because I would like to see the paper published.

The required revisions concern their response to Reviewers' 1 and 2 comment about the degree to which the methods they develop are novel and an improvement over other approaches. Therefore, I do not recommend publication if the authors include text that states directly or infers that their approach is an improvement over other methods in a general sense, and not specific to East Asia. Here is the authors' response to the novelty/added value issue raised by Reviewers 1 and 2:

“We agree with the reviewer that there are several studies using farmer field data and machine learning techniques in agronomy. However, the studies cited by the reviewer focused on descriptive analytics to generate population level insights. The approach developed in the current manuscript is a significant departure from those studies. It combines field-specific insights, solution targeting for single and multiple production constraints, and predictive analytics to assess the value of different rice development strategies in a spatial framework across multiple dimensions (yield, profitability, NUE). The proposed method is therefore new and adds value to existing studies on yield gap analysis using farmer field data.”

Reviewer comment: While I concur that the revised paper is new and adds value to existing approaches employed in East India, I am not convinced it adds value over other recently developed methods to evaluate yield gaps and yield constraints across spatial scales using large-n field data. Perhaps their method is an improvement over other approaches used to date in East India, but it has not been proven more effective than recently developed methods using equally large-n single field datasets in other major crop production areas, including data-poor and data-rich regions. If the authors wish to make a global claim to have developed a new approach “that adds value” for identifying field-level constraints to increased farm yields, they should publish such in a disciplinary journal with validation comparing effectiveness of their new approach with other recently developed approaches that have also used large-n datasets to do the same thing. For the paper under review here, I suspect the authors can rightly claim that the approach is an improvement over other methods that evaluate rice systems in East India. But it remains to be seen if it is an improvement over recently published approaches applied to major crops elsewhere. So, bottom line, if the authors wish claim their method is an improvement over other methods in a second revision, they must specify that any improvements are for like-studies in East India.

Additional comments below in response to the authors' more detailed rebuttal to this issue:

Point (1) from authors' rebuttal: "this is the first comprehensive assessment of sub-national rice yields in India where data-driven insights identify drivers of yield gaps in different production contexts (Figures 2 and 3). We note this is a very timely issue given the recent policies of the Government of India to temporarily ban rice exports over concerns about the adequacy of domestic supplies."

Reviewer response: Likely a true statement because it is limited to the region under study, namely, the East India region

Point (2) in authors' rebuttal: "This is the first comprehensive assessment of yield constraints in Eastern India, a priority region to the Indian government due to high dependency on rice and wheat farming and the largest density of population living below poverty line in the world. Please note that India is a big country and because there are studies in the states of Punjab and Haryana it doesn't mean such studies are relevant to Bihar, where farming systems and production constraints are different. Moreover, our analysis has direct policy relevance by placing results in a spatial framework, simultaneously addressing multiple production constraints, and evaluating management options across three sustainability dimensions. This policy relevance of our study was also highlighted by Reviewer 2."

Reviewer comment: True and appropriately limited to the East India region.

Point (3) from authors' rebuttal: "this is the first study using the SHAP methodology in agronomy, certainly so in the context of yield gap analysis with farmer field data. This allowed us to identify yield constraints for individual fields (Figure 3 and 4)."

Reviewer response: Not sure if this statement is true; perhaps it is? But it is only speculation unless the authors perform a comprehensive literature review.

Points (4) and (5) from authors' rebuttal: "Beyond estimating SHAP values with random forest, we also conducted ex-post analyses on those estimates to cluster fields based on their extent of water and nutrient co-limitation. To our knowledge, no other study has done this to date." And, "has a spatial dimension allowing us to unravel which yield constraints are prevalent where and to conduct an ex-ante quantification of aggregated benefits from practice change in a spatially explicit manner. Overall, these steps provide insights into which interventions should be prioritize where, and what are associated returns-on investment of doing so, towards increasing rice productivity in Eastern India."

Reviewer Comment: These statements is likely true for East Asia, but robust methods to cluster fields with regard to extent of water, nitrogen and some additional constraints have been published based on studies conducted on other crops and regions. Again, if the authors limit these statements to East India, then OK. Bottom line, any statements that state or imply general, global relevance must be revised to specify relevance for East India only.

Reviewer #3 (Remarks to the Author):

Thank you for the revisions and responses. I have no further comments.

#Response to reviewer's query (Responses are highlighted in green)

The authors should be commended for greatly improving their paper in a number of ways through revisions made in response to the three external reviews. I recommend publication pending minor, but important, revision. I would not recommend publication without making these revisions. I hope they make the appropriate revisions because I would like to see the paper published.

Response: We thank reviewer for acknowledging our efforts to improve the manuscript based on their constructive comments.

The required revisions concern their response to Reviewers' 1 and 2 comment about the degree to which the methods they develop are novel and an improvement over other approaches. Therefore, I do not recommend publication if the authors include text that states directly or infers that their approach is an improvement over other methods in a general sense, and not specific to East Asia.

Here is the authors' response to the novelty/added value issue raised by Reviewers 1 and 2:

“We agree with the reviewer that there are several studies using farmer field data and machine learning techniques in agronomy. However, the studies cited by the reviewer focused on descriptive analytics to generate population level insights. The approach developed in the current manuscript is a significant departure from those studies. It combines field-specific insights, solution targeting for single and multiple production constraints, and predictive analytics to assess the value of different rice development strategies in a spatial framework across multiple dimensions (yield, profitability, NUE). The proposed method is therefore new and adds value to existing studies on yield gap analysis using farmer field data.”

Response: The analysis we present is built on several linked methodologies, not a single method *per se*. Based on the reviewer's request, we have modified the introduction to state:

“Our approach leverages a spatially balanced sampling framework and interpretation of machine learning yield predictions with SHapley Additive exPlanations (SHAP) values. We then use the same models to investigate the yield intensification and sustainability impacts of changes in key agronomic practices through ex-ante scenario analysis with and without solution targeting.”

We have ensured that all adjacent studies are appropriately cited in the Introduction and Discussion sections and describe how our work adds value to prior studies:

“Nevertheless, insights into the nature of yield gaps in the emerging priority regions are generalized and incompletely understood. In this study, we used an analytics-based approach to fill this knowledge gap and identify context-dependent pathways for sustainable rice intensification. Our approach quantifies field-specific yield constraints, in contrast to earlier studies providing population level yield gap insights^{14,39,40}. This type of local interpretation of machine learning models can support intervention targeting to more effectively and efficiently narrow yield gaps in production fields that are likely to accrue the highest benefits.”

Reviewer comment: While I concur that the revised paper is new and adds value to existing approaches employed in East India, I am not convinced it adds value over other recently developed methods to evaluate yield gaps and yield constraints across spatial scales using large-n field data. Perhaps their method is an improvement over other approaches used to date in East India, but it has not been proven more effective than recently developed methods using equally large-n single field datasets in other major crop production areas, including data-poor and data-rich regions. If the authors wish to make a global claim to have developed a new approach “that adds value” for identifying field-level constraints to

increased farm yields, they should publish such in a disciplinary journal with validation comparing effectiveness of their new approach with other recently developed approaches that have also used large-n datasets to do the same thing.

Response: It is important to acknowledge that there are a diverse set of scientific objectives that underpin work on yield gaps and sustainable intensification. In some cases, the intent is only to quantify the magnitude of the yield gap and to identify or rank constraints. In our case, we focus on scenario analysis with predictive modelling to elucidate field-specific intensification pathways, their spatial distribution, the implications of solution targeting for single and multiple interventions from an integrative perspective (i.e, yield, economic, and ecosystem implications). As noted by the reviewer, methods intercomparison is indeed very important and should be a priority, but while recognizing that different methods (and combinations thereof...) serve different scientific purposes. To the best of our knowledge, we cite all relevant literature. While there are certainly commonalities in the approaches, our objectives are somewhat distinct from prior studies.

For the paper under review here, I suspect the authors can rightly claim that the approach is an improvement over other methods that evaluate rice systems in East India. But it remains to be seen if it is an improvement over recently published approaches applied to major crops elsewhere. So, bottom line, if the authors wish claim their method is an improvement over other methods in a second revision, they must specify that any improvements are for like-studies in East India.

Response: We have cited the relevant literature noted by the reviewer and make no claim to ‘improvement’ over other methods. As described above, the objectives of our study are adjacent but not the same to prior yield gap work.

Additional comments below in response to the authors’ more detailed rebuttal to this issue:

Point (1) from authors’ rebuttal: “this is the first comprehensive assessment of sub-national rice yields in India where data-driven insights identify drivers of yield gaps in different production contexts (Figures 2 and 3). We note this is a very timely issue given the recent policies of the Government of India to temporarily ban rice exports over concerns about the adequacy of domestic supplies.”

Reviewer response: Likely a true statement because it is limited to the region under study, namely, the East India region

Response: Thank you.

Point (2) in authors’ rebuttal: “This is the first comprehensive assessment of yield constraints in Eastern India, a priority region to the Indian government due to high dependency on rice and wheat farming and the largest density of population living below poverty line in the world. Please note that India is a big country and because there are studies in the states of Punjab and Haryana it doesn’t mean such studies are relevant to Bihar, where farming systems and production constraints are different. Moreover, our analysis has direct policy relevance by placing results in a spatial framework, simultaneously addressing multiple production constraints, and evaluating management options across three sustainability dimensions. This policy relevance of our study was also highlighted by Reviewer 2.”

Reviewer comment: True and appropriately limited to the East India region.

Response: Thank you.

Point (3) from authors’ rebuttal: “this is the first study using the SHAP methodology in agronomy,

certainly so in the context of yield gap analysis with farmer field data. This allowed us to identify yield constraints for individual fields (Figure 3 and 4).”

Reviewer response: Not sure if this statement is true; perhaps it is? But it is only speculation unless the authors perform a comprehensive literature review.

Response: With reference to using SHAP for local interpretation of predictions for yield gap analysis, we are happy to include additional references, but a systematic search of the literature did not reveal any omissions.

Points (4) and (5) from authors’ rebuttal: “Beyond estimating SHAP values with random forest, we also conducted ex-post analyses on those estimates to cluster fields based on their extent of water and nutrient co-limitation. To our knowledge, no other study has done this to date.” And, “has a spatial dimension allowing us to unravel which yield constraints are prevalent where and to conduct an ex-ante quantification of aggregated benefits from practice change in a spatially explicit manner. Overall, these steps provide insights into which interventions should be prioritize where, and what are associated returns-on investment of doing so, towards increasing rice productivity in Eastern India.”

Reviewer Comment: These statements is likely true for East Asia, but robust methods to cluster fields with regard to extent of water, nitrogen and some additional constraints have been published based on studies conducted on other crops and regions. Again, if the authors limit these statements to East India, then OK. Bottom line, any statements that state or imply general, global relevance must be revised to specify relevance for East India only.

Response: We have cited the below literature as suggested by the reviewer and the statements in our manuscript pertain to results **only for India**.

Rizzo, G., Agus, F., Batubara, S.F., Andrade, J.F., Edreira, J.I.R., Purwantomo, D.K., Anasiru, R.H., Marbun, O., Ningsih, R.D., Ratna, B.S. and Yulianti, V. A farmer data-driven approach for prioritization of agricultural research and development: A case study for intensive crop systems in the humid tropics. *Field Crops Research* **297**, p.108942 (2023).

Andrade, J.F., Mourtzinis, S., Edreira, J.I.R., Conley, S.P., Gaska, J., Kandel, H.J., Lindsey, L.E., Naeve, S., Nelson, S., Singh, M.P. and Thompson, L. Field validation of a farmer supplied data approach to close soybean yield gaps in the US North Central region. *Agricultural Systems* **200**, p.103434 (2022).

Reviewer #3 (Remarks to the Author):

Thank you for the revisions and responses. I have no further comments.

Response: Thank you.